# Genomic epidemiology of COVID-19 in care homes in the east of England

William L Hamilton[1,2†]*, Gerry Tonkin-Hill[3†], Emily R Smith[4], Dinesh Aggarwal[2,5], Charlotte J Houldcroft[6], Ben Warne[1,2], Luke W Meredith[6], Myra Hosmillo[6], Aminu S Jahun[6], Martin D Curran[7], Surendra Parmar[7], Laura G Caller[6,8], Sarah L Caddy[6], Fahad A Khokhar[2], Anna Yakovleva[6], Grant Hall[6], Theresa Feltwell[6], Malte L Pinckert[6], Iliana Georgana[6], Yasmin Chaudhry[6], Colin S Brown[5], Sonia Gonçalves[3], Roberto Amato[3], Ewan M Harrison[3], Nicholas M Brown[1,7], Mathew A Beale[3], Michael Spencer Chapman[3,9], David K Jackson[3], Ian Johnston[3], Alex Alderton[3], John Sillitoe[3], Cordelia Langford[3], Gordon Dougan[2], Sharon J Peacock[2], Dominic P Kwiatowski[3], Ian G Goodfellow[6], M Estee Torok[1,2]*, COVID-19 Genomics Consortium UK

[1]Cambridge University Hospitals NHS Foundation Trust, Departments of Infectious Diseases and Microbiology, Cambridge, United Kingdom; [2]University of Cambridge, Department of Medicine, Cambridge, United Kingdom; [3]Wellcome Sanger Institute, Hinxton, United Kingdom; [4]Cambridgeshire County Council, Cambridge, United Kingdom; [5]Public Health England, Colindale, United Kingdom; [6]University of Cambridge, Department of Pathology, Division of Virology, Cambridge, United Kingdom; [7]Public Health England Clinical Microbiology and Public Health Laboratory, Cambridge, United Kingdom; [8]The Francis Crick Institute, London, United Kingdom; [9]Department of Haematology, Hammersmith Hospital, Imperial College Healthcare NHS Trust, London, United Kingdom

*For correspondence:
will.l.hamilton@gmail.com (WLH);
et317@cam.ac.uk (MET)

†These authors contributed equally to this work

**Abstract** COVID-19 poses a major challenge to care homes, as SARS-CoV-2 is readily transmitted and causes disproportionately severe disease in older people. Here, 1167 residents from 337 care homes were identified from a dataset of 6600 COVID-19 cases from the East of England. Older age and being a care home resident were associated with increased mortality. SARS-CoV-2 genomes were available for 700 residents from 292 care homes. By integrating genomic and temporal data, 409 viral clusters within the 292 homes were identified, indicating two different patterns – outbreaks among care home residents and independent introductions with limited onward transmission. Approximately 70% of residents in the genomic analysis were admitted to hospital during the study, providing extensive opportunities for transmission between care homes and hospitals. Limiting viral transmission within care homes should be a key target for infection control to reduce COVID-19 mortality in this population.

## Introduction

Care homes are at high risk of experiencing outbreaks of SARS-CoV-2. COVID-19 is associated with higher mortality in older people and those with comorbidities including cardiovascular and respiratory disease (*Williamson et al., 2020*), making the care home population especially vulnerable. As of week ending 30th June 2020, the UK Office for National Statistics (ONS) estimated that 30.2% of all deaths due to COVID-19 (13,417 deaths) in England occurred in care homes, and 63.9% (28,390 deaths) occurred in hospital (*Office for National Statistics, 2020a*). Most of the COVID-19 deaths in hospital were in persons aged 65 years and over (86.1%). Deaths due to confirmed COVID-19

from this period may be underestimates due to limitations on diagnostic testing; the ONS estimates that from 28 December 2019 to 12 June 2020, there were 29,393 excess deaths in care homes compared to the expected number based on previous years, of which only two thirds are explained by recorded COVID-19 (*Office for National Statistics, 2020b*). To date, SARS-CoV-2 transmission in care homes has not been systematically studied with linkage of epidemiological and genomic data on a large scale.

Care homes are defined by the Care Quality Commission (CQC), the independent regulator of adult health and social care in England, as 'places where personal care and accommodation are provided together' (*Care Quality Commission, 2020a*). In 2011, 291,000 people aged 65 or older were living in care homes in England and Wales, representing 3.2% of the total population at this age; 82.5% of the care home population was aged 65 years or older (*Office for National Statistics, 2014*). Care homes are known to be high-risk settings for infectious diseases, owing to a combination of the underlying vulnerability of residents who are often frail and elderly with multiple comorbidities, the shared living environment with multiple communal spaces, and the high number of interpersonal contacts between residents, staff, and visitors in an enclosed space (*Curran, 2017*; *Lansbury et al., 2017*; *Strausbaugh et al., 2003*). Understanding the transmission dynamics of SARS-CoV-2 within care homes is therefore an urgent public health priority.

Rapid SARS-CoV-2 sequencing combined with detailed epidemiological analysis has been used to trace viral transmission networks in hospital and community-based healthcare settings (*Meredith et al., 2020*). This study was based in Cambridge University Hospitals (CUH), a secondary care provider and tertiary referral centre in the East of England, UK. The study focused on identifying hospital-acquired and healthcare-associated infections by integrating genomic and epidemiological data with hospital Infection Prevention and Control (IPC) systems. While clusters involving care home residents and healthcare workers were observed, the study was not intended to analyse care home transmission specifically and focused on samples tested at CUH to provide information for IPC on potentially hospital-acquired infections. Previous epidemiological studies of COVID-19 specifically in care homes have been limited in population size, temporal scale and/or the amount of genomic data included (*Arons et al., 2020*; *Burton et al., 2020*; *Graham et al., 2020*; *Kemenesi et al., 2020*; *Quicke et al., 2020*). Here, genomic epidemiology is used to investigate viral transmission dynamics in care home residents across the East of England (EoE), the fourth largest of the nine official regions in England (*Office for National Statistics, 2011*). Several key questions of public health concern are addressed: What is the burden of care-home-associated COVID-19 tested in the region? What are the outcomes for care home residents admitted to hospital with COVID-19? Does SARS-CoV-2 spread between care home residents from the same care home via a single introduction and subsequent transmission, or through multiple independent acquisitions of the virus among residents? Finally, is there evidence of viral transmission between care homes and hospitals?

## Results

### COVID-19 case numbers from care home and non-care home residents included in the study

A total of 7,406 SARS-CoV-2 positive samples from 6600 individuals were identified in the study period (26th February to 10th May 2020) (*Figure 1*), and care home residency status was determined in 6413 (*Figure 1—figure supplement 1*) – the remaining 187 cases had missing address data and care home status could not be determined. The samples were tested at the Public Health England (PHE) Clinical Microbiology and Public Health Laboratory (CMPHL) in Cambridge, which receives samples from across the East of England (EoE). Positive cases came from 37 submitting organisations including regional hospital laboratories and community-based testing services (Supplementary Materials). The proportion of samples coming from different sources changed over the study period (*Figure 1—figure supplement 2*). This likely reflects a combination of regional hospitals establishing their own testing facilities, increasing availability of community testing in the UK, and the implementation of national policies that increased the scope of care home testing (*Figure 1—figure supplement 3*). Overall, the study population included almost half of the COVID-19 cases diagnosed in the EoE at this time (*Public Health England, 2020a*), with the remainder being tested at other laboratory sites.

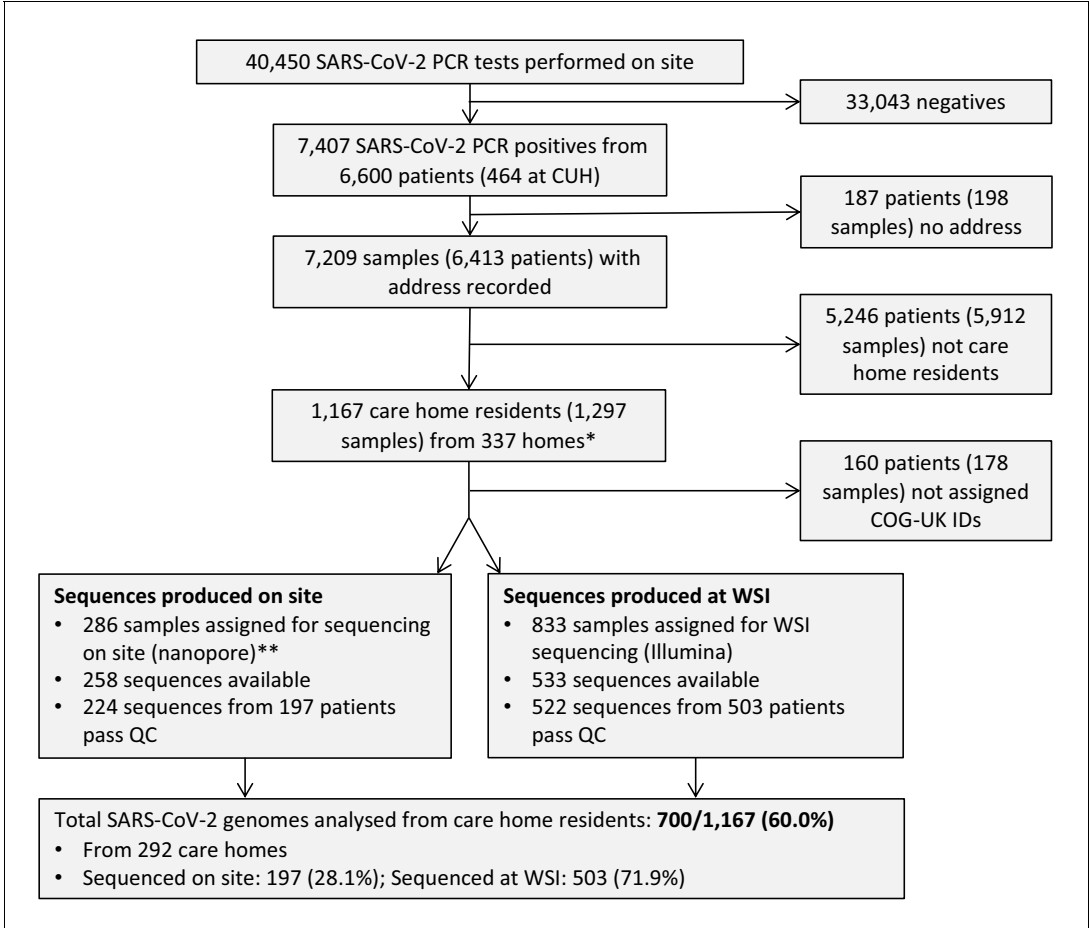

**Figure 1.** Study flow diagram Out of 6600 patients testing positive in the Cambridge Microbiology Public Health Laboratory (CMPHL) during the study period, 1167 were identified as being care home residents from 337 care homes. (The methodology for assigning care home status is described in main text and *Figure 1—figure supplement 1*). Out of 1297 samples from 1167 care home residents, 286 samples were assigned for nanopore sequencing on site and 833 samples for sequencing at the Wellcome Sanger Institute (WSI). Of these, 258 and 533 sequences were available and downloaded from the MRC-CLIMB server at the time of running the analysis, respectively. Of these available genomes, 224 and 522 passed sequencing quality control thresholds (described in Materials and methods), respectively. This yielded the final analysis set of 700 high-coverage genomes from care home residents (representing 292 care homes): 197 genomes sequenced on site by nanopore and 503 sequences at WSI by Illumina. * 193 care homes were registered with the CQC as being residential homes without nursing care, referred to as 'residential homes' in main text, and 144 had nursing care available, referred to as 'nursing homes'. ** Samples were selected for nanopore sequencing on site if they were inpatients or healthcare workers at Cambridge University Hospitals NHS Foundation Trust (CUH), where we prioritised rapid turnaround time to investigate hospital-acquired infections, plus a randomised selection of other East of England samples to provide broader genomic context to the CUH cases. The remaining samples not selected for nanopore sequencing on site, where available, were sent to WSI for sequencing.

The online version of this article includes the following figure supplement(s) for figure 1:

**Figure supplement 1.** Flow diagram for identifying care homes from Cambridge-COGUK metadata Steps for identifying care home residents (further details in Materials and methods).

**Figure supplement 2.** Breakdown of main organisations submitting samples to Cambridge PHE Laboratory over study period per week.

**Figure supplement 3.** UK care home testing policy timeline.

Of the study population, 1167/6413 (18.2%) were identified as care home residents from 337 care homes. 193/337 (57.3%) care homes were residential homes and 144/337 (42.7%) were nursing homes, with the majority located in five counties across EoE: Essex, Hertfordshire, Bedfordshire, Suffolk and Cambridgeshire (*Figure 2*). This represents around half of the care homes in the East of England which had reported suspected or confirmed COVID-19 outbreaks to PHE as of 11th May 2020 (*UK government, 2020a*). As expected, care home residents were older than non-care home residents (median age 86 years versus 65 years, respectively [$p < 10^{-5}$, Wilcoxon rank sum test]) (*Table 1*). There was a median of two cases per care home (range 1–22), with a highly skewed

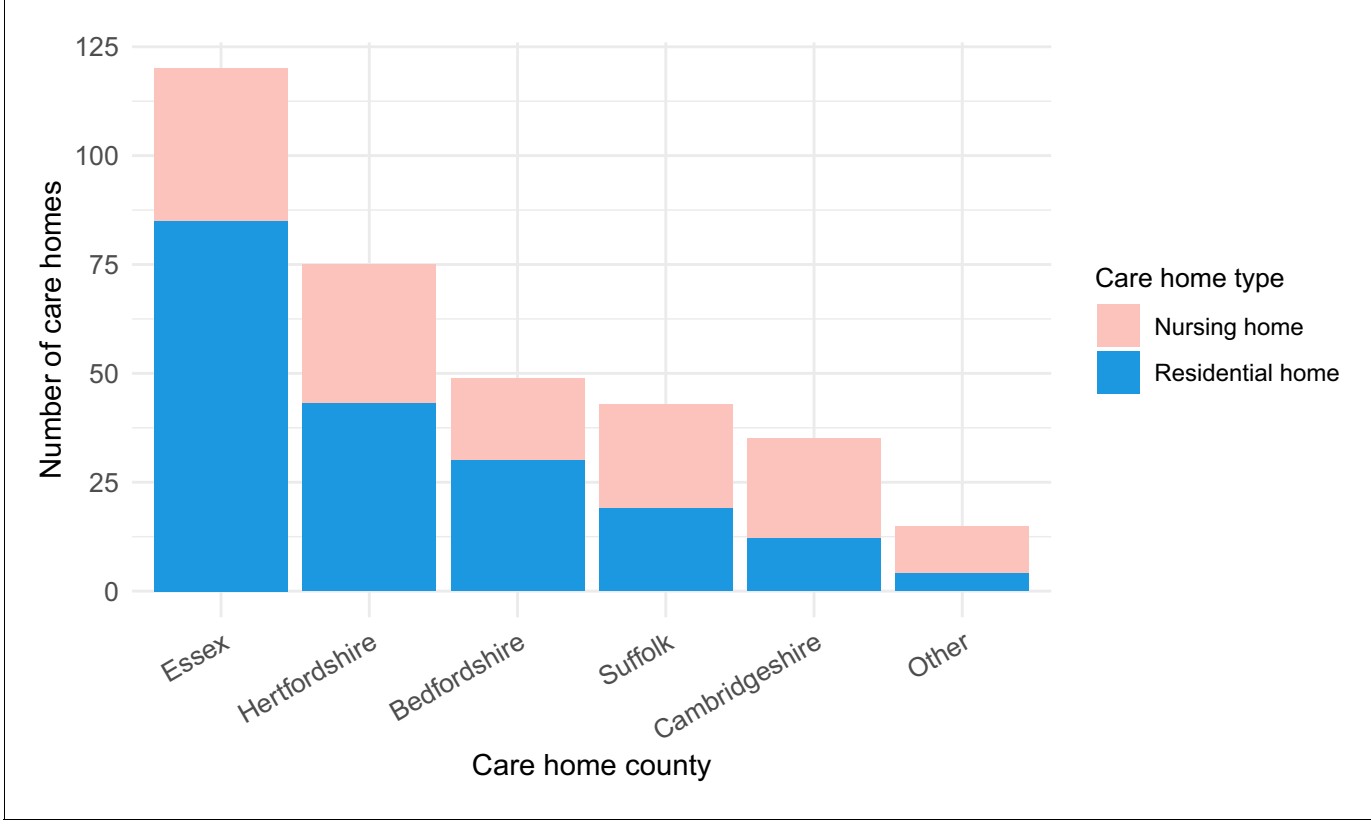

**Figure 2.** Care home locations by county, showing nursing, and residential homes. Only showing the five counties with the largest number of cases (all >25) to preserve patient anonymity. Definitions of 'nursing home' and 'residential home' are based on Care Quality Commission (CQC) information on whether nursing care is or is not present. If no nursing care is available the home is classified as a residential home. If the care home offers nursing care (including if it can offer both nursing and residential care) then the home is classified as a nursing home.
The online version of this article includes the following figure supplement(s) for figure 2:

**Figure supplement 1.** Distribution of cases per care home.

distribution: the 10 care homes (top 3%) with the largest number of cases contained 164/1167 (14.1%) of all care home cases (*Figure 2—figure supplement 1*).

The epidemic curve for all cases tested at the Cambridge CMPHL peaked in the end of March and early April (*Figure 3*). Care home residents comprised a greater proportion of cases in late April and May than in March (*Figure 3A*, *Table 2*). This may reflect the changing profile of samples submitted to the CMPHL, as more regional hospitals had their own testing capacity and a greater number of samples were submitted from community testing organisations in later weeks. However, a similar trend was observed for patients tested at Cambridge University Hospitals, with the proportion of community-onset care home-associated cases increasing from <5% in March to a peak of 14/49 (28.6%) in mid-April (*Figure 3B*, *Table 3*). This may suggest that transmission involving care home residents took longer to decline following national lockdown (implemented on 23rd March 2020 in the UK) than transmission in the non-care home general community.

## Mortality of COVID-19 infections for care home and non-care home residents tested in hospital

Of 6600, 464 (7%) individuals with positive COVID-19 tests were patients tested at Cambridge University Hospitals. Richer metadata were available for this subset of patients via the hospital electronic records system. Seventy-two of 464 (15.5%) COVID-19 patients diagnosed at CUH were identified as care home residents (*Table 1*, *Figure 3B*), of which < 7% were admitted to the intensive care unit (ICU) and 34/72 (47.2%) died within 30 days of their first positive test (precise values not

**Table 1.** Epidemiological characteristics of care home and non-care home residents with COVID-19 included in the study.
The total sample set for this study comprised 6600 individuals. Of these, care home residency status could be established for 6413 (97.2%). 1167/6413 (18.2%) individuals were identified as being care home residents, of which 700/1167 (60.0%) had genomic data available that passed quality control filtering and were used for identifying care home clusters using the *transcluster* algorithm (described in Methods and main text). The subset of individuals (464/6600, 7.03%) that were tested at Cambridge University Hospitals (CUH) had richer metadata available and were used for analysing intensive care unit (ICU) admissions and 30 day mortality after first positive test, shown here. Not showing precise values where the number of cases is equal to or less than five individuals, to preserve patient anonymity. Ct = Cycle threshold; CUH = Cambridge University Hospitals; ICU = Intensive Care Unit; IQR = interquartile range.

| Variable | Care home residents (all) | Non-care home residents (all) | Care home residents with genomes |
|---|---|---|---|
| Number (%) | 1167/6413 (18.2%) | 5246/6413 (81.8%) | 700/1167 (60%) |
| Female (%) | 624/1167 (53.5%) | 2338/5246 (44.6%) | 363/700 (51.9%) |
| Male (%) | 543/1167 (46.5%) | 2908/5246 (55.4%) | 337/700 (48.1%) |
| Age in years (median, IQR, range) | 86 (IQR: 79–90, range: 30–100) | 65 (IQR: 48–80, range: 0–100) | 86 (IQR: 78–90, range: 42–99) |
| Diagnostic Ct value | 26 (IQR: 22–29) | 25 (IQR: 21–29) | 24 (IQR: 20–27) |
| Tested at CUH (%) | 72/464 (15.5%) | 392/464 (84.5%) | 54/72 (75%) |
| CUH patient admitted to ICU (%) | <5/72 (<7%) | 84/392 (21.4%) | <5/54 (<9%) |
| CUH patient 30 day mortality (%) | 34/72 (47.2%) | 78/392 (19.9%) | 23/54 (42.6%) |
| Number of care homes | 337 | - | 292 |
| Cases/ care home (median, IQR, range) | 2 (IQR: 1–5, range: 1–22) | - | 2 (IQR: 1–3, range: 1–18) |
| Care homes with ≥ 5 cases | 85/337 (25.2%) | - | 32/292 (11%) |

shown where the number of individuals is equal to or below five, to protect patient anonymity). In comparison, amongst non-care home residents, 84/392 (21.4%) were admitted to the ICU and 78/392 (19.9%) died within 30 days of diagnosis. In a logistic regression analysis, older age, care home residency, ICU admission, and lower diagnostic cycle threshold (Ct) values were associated with increased odds of mortality at 30 days from diagnosis (*Figure 4*, *Table 4*). The odds of mortality within 30 days of diagnosis did not differ between residents at nursing homes versus residential homes in a separate logistic regression analysis.

## Identifying viral clusters within care homes using genomic and epidemiological data

Genome sequence data were available for 700/1167 (60.0%) care home residents from 292 care homes (*Figure 3—figure supplement 1*). There was a median of eight single-nucleotide polymorphisms (SNPs) separating care home genomes, compared to nine for randomly selected non-care home samples (p=0.95, Wilcoxon rank sum test) (*Figure 5—figure supplement 2*), similar to the EoE region described previously (*Meredith et al., 2020*). The proportion of viral lineage B.1.1 increased over the study period in both care home residents and non-care home residents (*Figure 5*, *Table 5*), consistent with European trends (*Alm et al., 2020*). With ongoing viral evolution, descendent lineages of B.1 and B.1.1 also rose in frequency and were commonly found in England during the relevant time period. This suggests that the SARS-CoV-2 lineages circulating in care homes were similar to those found across the EoE outside of care homes. Consistent with this, care home and non-care home samples were intermixed across the phylogenetic tree (*Figure 6A*), suggesting viral transmission could pass between care homes and non-care home settings. No new viral lineages from outside the UK were observed, which may reflect the success of travel restrictions in limiting introductions of new lineages into the general population.

The 10 care homes with the largest number of genomes (top ~3%) contained 102/700 (14.6%) of all samples with genomic data available. For several of these 10 care homes, all cases clustered closely together on a phylogenetic tree with zero or one pairwise SNP differences, consistent with a single 'outbreak' spreading within the care home (where an outbreak is defined as two or more cases linked in time or place *McAuslane and Morgan, 2014*; *Figure 6* and *Figure 6—figure supplement 1*). By contrast, several care homes were 'polyphyletic', with cases distributed across the

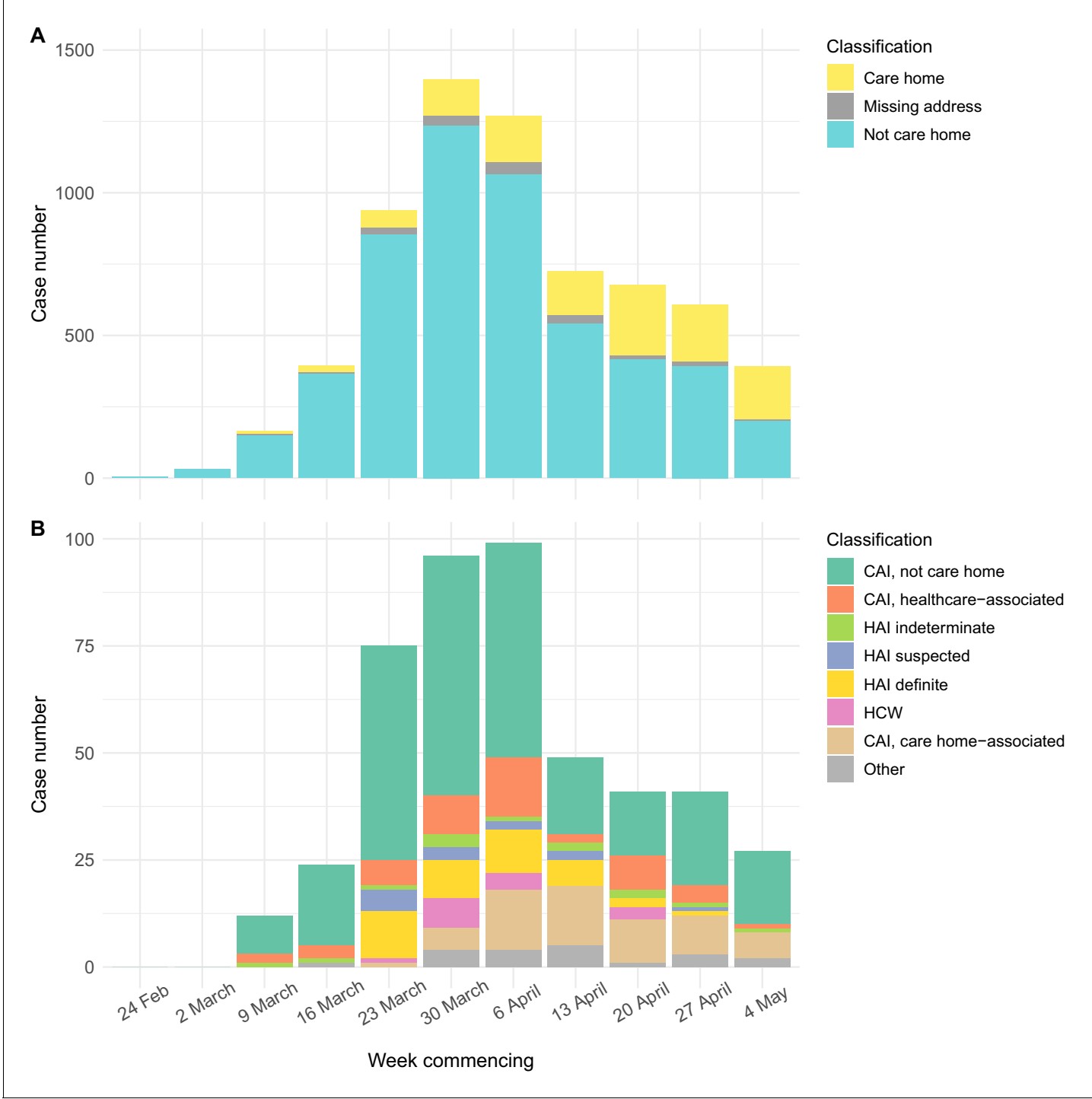

**Figure 3.** Epidemic curves for EoE and CUH showing care home residents. Number of positive cases per week over the study period for different infection sources, for all samples tested from EoE at the Cambridge PHE laboratory (**A**), or those tested at CUH acute medical services (**B**). Peak of the epidemic for samples tested at the Cambridge PHE laboratory and CUH acute medical services were weeks commencing 30th March and 6th April, respectively. UK lockdown started 23rd March 2020. In both settings, a prolonged right-hand 'tail' was observed as case numbers gradually fell. The relative proportion of cases admitted from care homes increased over this period for both sample sets, while the contribution of general community cases fell more quickly. However, interpreting these trends is confounded by the changing profile of COVID-19 testing nationally and regionally. If the patient address was missing, and they were not a HCW, then the care home status was undetermined. CAI = Community Acquired Infection; EoE = East of England; HAI = Hospital Acquired Infection; HCW = Healthcare Worker; 'Other' mainly comprise inpatient transfers from other hospitals to CUH for which metadata was lacking to determine the infection category. CAI was considered 'healthcare-associated' if there had been healthcare contact

*Figure 3 continued on next page*

*Figure 3 continued*

within 14 days of first positive swab. The three categories of HAI were defined based on the difference in days between admission and first positive swab, reflecting increasing likelihood of hospital acquisition: indeterminate = 3–6 days; suspected 7–14 days; definite >14 days (as used in *Meredith et al., 2020*).

The online version of this article includes the following figure supplement(s) for figure 3:

**Figure supplement 1.** Care home residents per week showing genome sequencing site.

phylogenetic tree and higher pairwise SNP difference counts between samples, consistent with multiple independent introductions of the virus among residents.

The probability of two cases having linked transmission in an epidemiologically meaningful timeframe (for example direct transmission or within one or two intermediate hosts – likely the maximum practical limit for investigating the source of infection for a positive case) is a function of several factors. These include the pairwise genetic differences between viruses and their phylogenetic relatedness, the time difference between cases, and the opportunities for infection between people (for example, the frequency, duration and extent of close contact). For this continuous probability distribution, a pragmatic cut-off was used of ≥15% likelihood that samples were connected by ≤2 intermediate hosts, using a previously published algorithm called *transcluster* (*Stimson et al., 2019*), adjusted for SARS-CoV-2 (Materials and methods). Each care home was considered as a separate microcosm of transmission and the number of viral clusters per care home was estimated, with separate clusters implying distinct acquisition events among residents.

This clustering method identified 409 transmission clusters from 292 care homes (median one cluster per care home, range 1–4). Within each cluster, 673/775 (86.8%) of pairwise links had zero or one pairwise SNP differences (maximum 4), and 756/775 (97.5%) were sampled <14 days apart (maximum 22 days) (*Figure 7—figure supplement 4–5*). Clusters had a smaller distribution of sampling dates than for the total cases within each care home, as expected (*Figure 7—figure supplement 6*). For the 170/292 (58%) care homes with two or more cases with genomic data (578 individuals), there was a median of 9 (IQR: 4–15) days from the first case to the last case within each care home, up to a maximum of 50 days. In contrast, more clusters comprised only a single individual than for care homes, and for the 133/409 (33%) clusters with two or more cases with genomic data (424 individuals), there was a median of 5 (IQR: 1–11) days from the first case to the last case within each cluster, up to a maximum of 22 days ($p<10^{-5}$, Wilcoxon rank sum test comparing date differences for care homes vs clusters with two or more samples; comparison shown in *Figure 7—figure supplement 6*). The median and interquartile range for pairwise date differences between all

**Table 2.** Case numbers from care homes and non-care home residents per week for full dataset tested at Cambridge CMPHL.

Data plotted in *Figure 3A* of main text, showing case numbers for care homes, non-care homes, and undetermined, for all EoE samples tested at CMPHL. The proportion of COVID-19 cases from care home residents increased in April and May; however, this may reflect the changing profile of samples submitted to the Cambridge CMPHL rather than underlying epidemiological trends.

| Week commencing | Care home resident | Not determined | Not care home resident | Weekly total | Care home resident (%) |
|---|---|---|---|---|---|
| 24-Feb | 0 | 0 | ≤5 | ≤5 | 0.0% |
| 02-Mar | 0 | 0 | 31 | 31 | 0.0% |
| 09-Mar | 10 | 6 | 149 | 165 | 6.1% |
| 16-Mar | 25 | 6 | 364 | 395 | 6.3% |
| 23-Mar | 60 | 26 | 852 | 938 | 6.4% |
| 30-Mar | 126 | 35 | 1235 | 1396 | 9.0% |
| 06-Apr | 162 | 43 | 1064 | 1269 | 12.8% |
| 13-Apr | 154 | 31 | 540 | 725 | 21.2% |
| 20-Apr | 247 | 16 | 415 | 678 | 36.4% |
| 27-Apr | 198 | 16 | 393 | 607 | 32.6% |
| 04-May | 185 | 8 | 199 | 392 | 47.2% |

**Table 3.** Proportion of community acquired, care home-associated COVID-19 infections tested at Cambridge University Hospitals.

The proportion of community onset, care home-associated COVID-19 infections tested at Cambridge University Hospitals (CUH) peaked in mid to late April. Total cases shows the total number of new COVID-19 cases diagnosed at CUH that week. 'Community acquired' was defined as first positive test ≤48 hr from admission and no healthcare contact within the previous 14 days. Not showing precise values if number of patients is less than or equal to five to preserve patient anonymity.

| Week | Total weekly COVID-19 cases | Community acquired, care home-associated (%) |
| --- | --- | --- |
| 09-Mar | 12 | 0 (0%) |
| 16-Mar | 24 | 0 (0%) |
| 23-Mar | 75 | ≤5 (<7%) |
| 30-Mar | 96 | ≤5 (≤5.2%) |
| 06-Apr | 99 | 14 (14.1%) |
| 13-Apr | 49 | 14 (28.6%) |
| 20-Apr | 41 | 10 (24.4%) |
| 27-Apr | 41 | 9 (22.0%) |
| 04-May | 27 | 6 (22.2%) |

samples within each cluster is shown in *Figure 7—figure supplement 7*, and the date ranges for all care homes and clusters is in Supplementary Materials.

Transmission networks for the ten care homes with the largest number of genomes are shown in *Figure 7A*, indicating linked transmission clusters among residents based on the model assumptions and probability threshold (full dataset shown in *Figure 7—figure supplement 1*). Consistent with the phylogeny shown in *Figure 6A*, some care homes contained a single transmission cluster involving multiple cases (e.g. CARE0314), while others comprised multiple independent clusters (e.g. CARE0061) (*Table 6*). While care homes frequently had more than one introduction of the virus among residents (i.e. >1 cluster), there was typically a single dominant cluster responsible for the majority of cases within each care home. Of the 170 care homes with two or more residents with genomic data (comprising 578/700 (82.6%) care home residents with genomic data), 111/170 (65.3%) had a dominant cluster responsible for >50% of all cases in the care home. This rises to 74/90 (82.2%) of care homes with three or more residents with genomic data.

The contribution made by genomic data in defining care home clusters was quantified. Without genomic data (or access to more detailed epidemiology such as accommodation sub-structuring within care homes), clustering can only be based on temporal differences between cases. For example, if two groups of COVID-19 cases occur several months apart within a care home they could be inferred to have resulted from (at least) two separate introductions. However, this method cannot account for multiple introductions occurring around the same time, as may happen when community transmission is high. To quantify the impact made by adding genomic data, which can distinguish between genetically dissimilar viruses introduced at similar times, the *transcluster* algorithm was repeated using the same parameters as for the main analysis but assuming all genomes were identical. This yielded 316 clusters – 23% fewer than the 409 clusters yielded when incorporating genomics. This suggests that genomics makes a significant contribution to defining viral clusters; without genomic data, cluster sizes may be over-estimated and the number of separate viral introductions under-estimated. This is illustrated by care home CARE0263, in which all 12 residents tested positive within 3 days of each-other, but these are divided into three separate clusters by the *transcluster* algorithm (one dominant cluster of nine cases, one cluster of two cases and a single separate case (*Table 6*)); this is consistent with the phylogeny shown in *Figure 6A*, with samples split into three branches along the tree. Without genomic data, the three clusters in CARE0263 would have been impossible to distinguish.

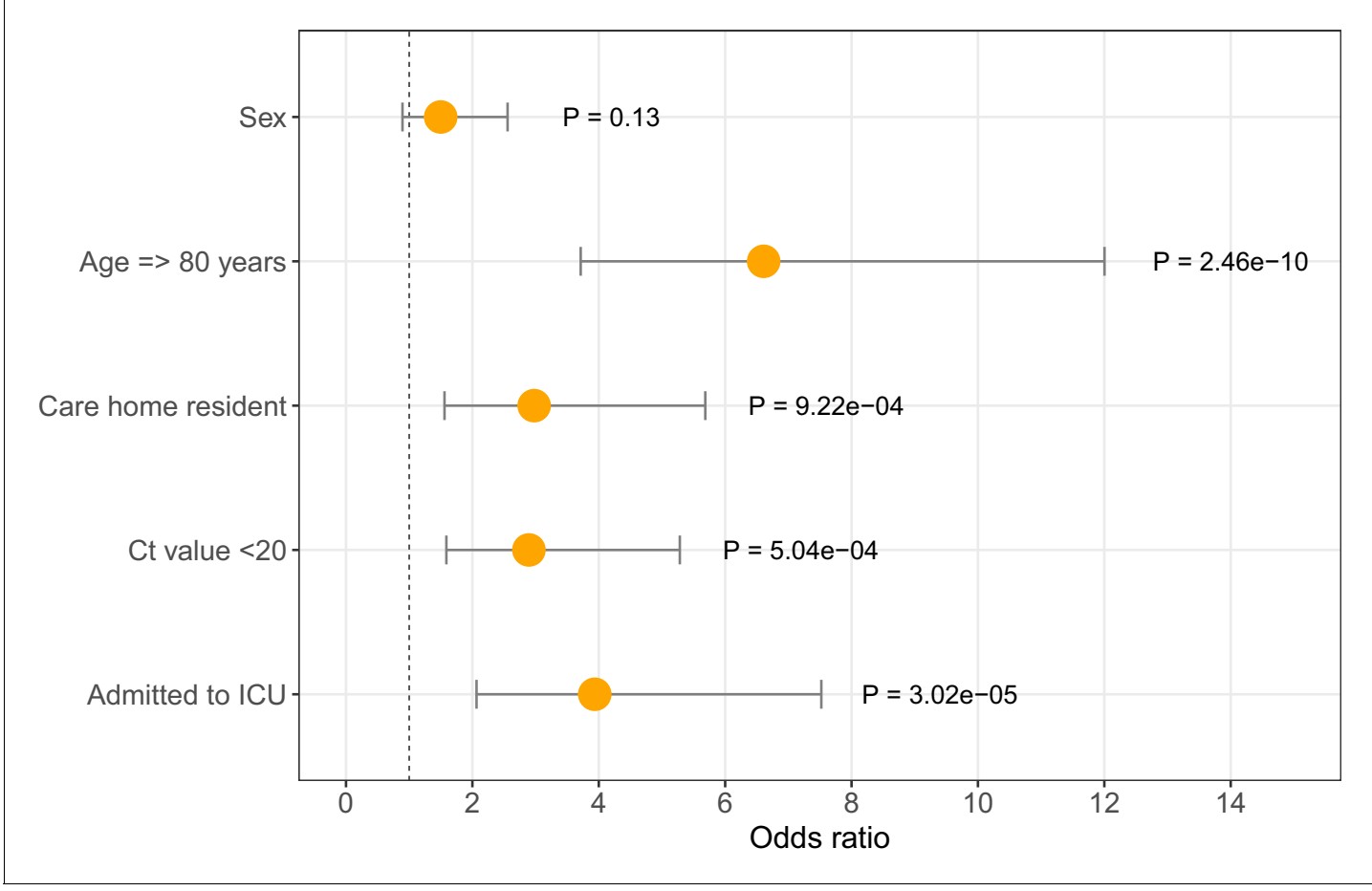

**Figure 4.** Odds ratios for mortality at 30 days. Logistic regression analysis showing odds of death at 30 days (with 95% confidence intervals) for five available metadata variables: patient sex, age (here categorised as ≥80 years), whether they were a care home resident, the diagnostic Ct value (here categorised as <20), and whether they were admitted to the intensive care unit. Overall there were 116 deaths within 30 days of diagnosis (out of 464 CUH patients). ICU = intensive care unit. Ct = Cycle threshold for diagnostic PCR.

The online version of this article includes the following figure supplement(s) for figure 4:

**Figure supplement 1.** Pairwise comparisons of mortality at 30 days, age and whether the person was a care home resident.

## Links between care homes and hospitals

Links between care homes and hospitals were investigated for the 700 care home residents with genomic data available. Of 700, 694 (99%) care home residents with genomic data had NHS

**Table 4.** Odds ratios for mortality at 30 days.

Logistic regression analysis of odds of mortality at 30 days. Age ≥ 80 years, being a care home resident, being admitted to ICU and Ct <20 were significantly associated with increased odds of death at 30 days post-diagnosis (p<0.05). OR = Odds Ratios. CI = Confidence Interval. ICU = intensive care unit. Ct = Cycle threshold for diagnostic PCR.

| Variable | OR | 95% CI low | 95% CI high | *P* value |
|---|---|---|---|---|
| Age >= 80 | 6.6 | 3.7 | 12.0 | 2.46E-10 |
| Sex | 1.5 | 0.9 | 2.6 | 1.30E-01 |
| Care resident status | 3.0 | 1.6 | 5.7 | 9.22E-04 |
| ICU admission | 3.9 | 2.1 | 7.5 | 3.02E-05 |
| Ct value < 20 | 2.9 | 1.6 | 5.3 | 5.04E-04 |

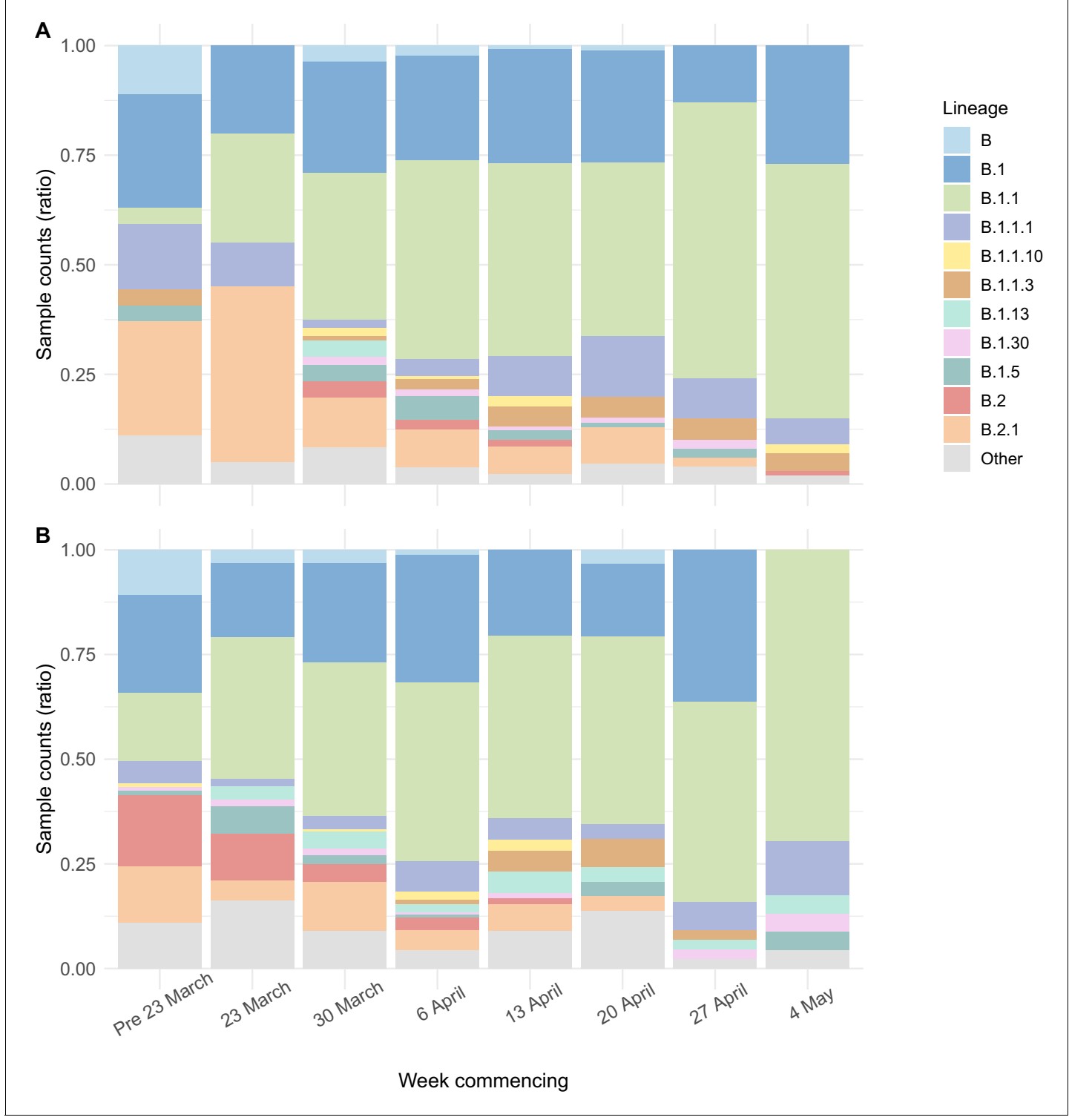

**Figure 5.** Viral lineage compositions in care home and non-care home samples. Plots showing the ratios of SARS-CoV-2 viral lineages for 700 care home resident genomes (**A**) and a randomly selected subset of 700 non-care home residents (**B**). The proportion of lineage B.1.1 increased over the study period in both care home and non-care home residents. Lineages defined using *pangolin*. Data also presented in *Table 5*.

The online version of this article includes the following figure supplement(s) for figure 5:

**Figure supplement 1.** Viral lineage compositions in care home and non-care home samples by count.

**Figure supplement 2.** Distribution of pairwise SNP differences between care home samples.

**Table 5.** Proportion of care home and non-care home samples that were lineage B.1.1.
The proportion of lineage B.1.1 (defined using the Pangolin tool) increased from earlier to later sampling weeks, for both care home and non-care home samples. Data based on the 700 care home residents with genomic data available and 700 randomly selected non-care home samples. 'Early' was defined as the period from the start of the study (26th February 2020) to 29th March 2020. 'Late' was defined as 20th April 2020 to the end of the study (10th May 2020).

| Care home status | Early | Late | % change |
|---|---|---|---|
| Care home resident | 6/47 (12.8%) | 155/286 (54.2%) | + 41.40% |
| Not care home resident | 39/173 (22.5%) | 50/96 (52.1%) | + 29.50% |

numbers available, which were linked to national hospital admissions data (Materials and methods) (*Table 7*). Of 694, 470 (67.7%) care home residents had at least one hospital admission within the study period, and 398/694 (57.3%) were deemed to have been admitted to hospital with COVID-19 (i.e. their first positive sample was taken within 2 days prior to admission up to 7 days post-admission). Forty of 694 (5.8%) cases were categorised as suspected hospital-acquired COVID-19 infections, defined as first positive test being 7 days or more after their hospital admission date and prior to their discharge date (N = 13) or within 7 days following their hospital discharge (N = 27) (*Table 7*). Of 694, 230 (33.1%) individuals were discharged from hospital within 7 days of their first positive test, and thus could potentially have been infectious at the time of hospital discharge (*Byrne et al., 2020*).

## Viral clusters linking care home residents and healthcare workers

Potential transmission networks involving care home residents and healthcare workers (HCW) were investigated for people tested at CUH (HCW data were not available outside of CUH). This analysis comprised 54 care home residents tested at CUH and 76 HCW with genomic data available. Clusters were defined using the same method as for the care home resident analysis (described above), but allowing HCW to belong to clusters from multiple care homes, so residents from several care homes could be linked to the same HCW. 38/54 (70.4%) care home residents had possible links with HCW using this relaxed threshold. However, on review of the medical records we could only identify strong epidemiological links for 14/54 (26.0%) residents from two care home clusters, CARE0063 and CARE0114. The CARE0063 cluster has been described previously (*Meredith et al., 2020*) and includes care home residents, a carer from that same care home and another from an unknown care home, paramedics and people living with the above. The CARE0114 cluster comprises several care home residents and acute medical staff working at CUH who cared for at least one of the residents. The *transcluster* method does not assign probabilities for directionality of transmission and cannot determine precise person-to-person transmission chains. While all residents from a care home cluster may link to a given HCW, in reality the resident-HCW transmission event may have only involved one of the residents from that cluster, so the proportion of residents with links to HCW may be inflated. Nonetheless, these data show that two care home clusters involved HCW, one based mainly in the community and the other with hospital-based staff at CUH.

Residents from a third care home, CARE0273, also had strong transmission links to the paramedics and carers involved in the CARE0063 cluster. These two care homes are within 1 km of eachother and the cases cluster together on the phylogenetic tree, raising the possibility of shared transmission between them. A plausible transmission network connecting the residents at these two care homes and the shared HCWs could be made with at most zero SNPs and 3 days between sampled cases (*Figure 7B*); these links are in the top 1.1% of all pairwise transmission probabilities inferred using the *transcluster* algorithm. However, without confirmatory epidemiological data this interpretation remains speculative.

## Discussion

The genomic epidemiology of SARS-CoV-2 in care homes in the East of England was investigated. Care home residents comprised a large fraction of COVID-19 diagnoses in the 'first wave' of the

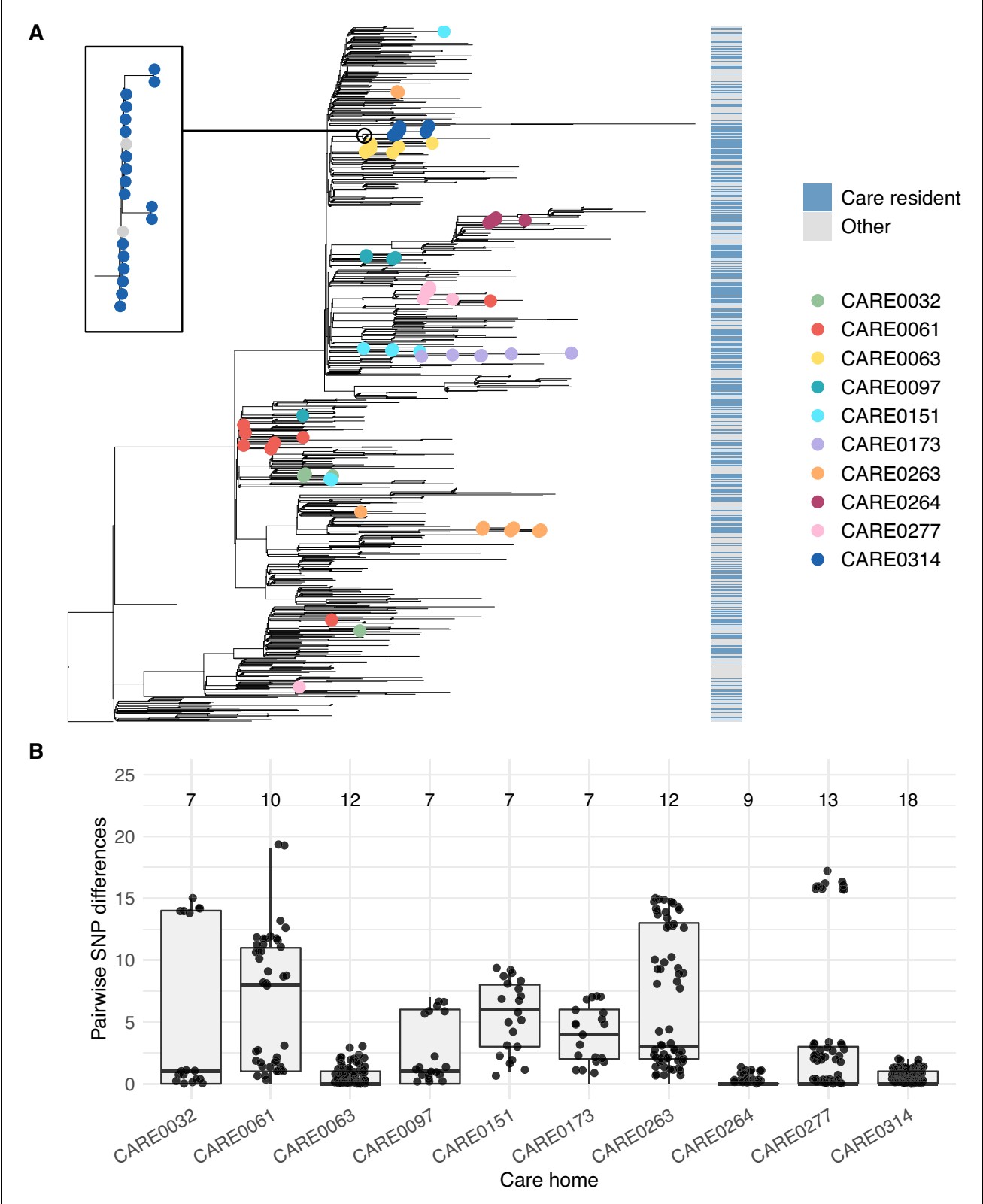

**Figure 6.** Care home clustering on viral phylogenetic tree and within-care home pairwise SNP differences. (**A**) Phylogenetic tree of 1400 East of England SARS-CoV-2 genomes rooted on a sample from Wuhan, China, collected December 2019, including 700 care home residents and 700 randomly selected non-care home residents. The colour bar (right) indicates whether samples were from care home residents (blue) or non-care home residents (grey). Samples from the 10 care homes with the largest number of genomes are highlighted by coloured circles on branch tips. A magnified

*Figure 6 continued on next page*

*Figure 6 continued*

subtree of the branch containing all 18 samples from care home CARE0314 is shown to the left. These genomes were all either identical or differed by one SNP from the most common genome in this cluster. Two non-care home genomes are also present in this group. Across the dataset, viruses from care home residents and people not living in care homes are phylogenetically intermixed, consistent with viral transmission between these two settings. (**B**) Distributions of pairwise SNP differences for the 10 care homes with the largest number of genomes (same samples as highlighted in the branch tips of panel A). Numbers above each box indicate the number of genomes present from that care home. Among the ten care homes with the largest number of genomes, some clustered closely on the phylogenetic tree with low pairwise SNP differences (e.g. CARE0063, CARE0264, CARE0314); in contrast, some care homes were distributed across the tree with higher pairwise SNP differences (e.g. CARE0061, CARE0151, CARE0173, CARE0263). Clusters within each care home were defined using integrated genomic and temporal data using the *transcluster* algorithm and are shown in ***Figure 7***.

The online version of this article includes the following figure supplement(s) for figure 6:

**Figure supplement 1.** Phylogenetic tree of all available genomes highlighting care home and non-care home samples.

pandemic in this region: up to a quarter of patients in the peak weeks of late March and early April tested at CUH were admitted from care homes. Older age and being from a care home were correlated with each other and were both associated with significantly increased odds of mortality within

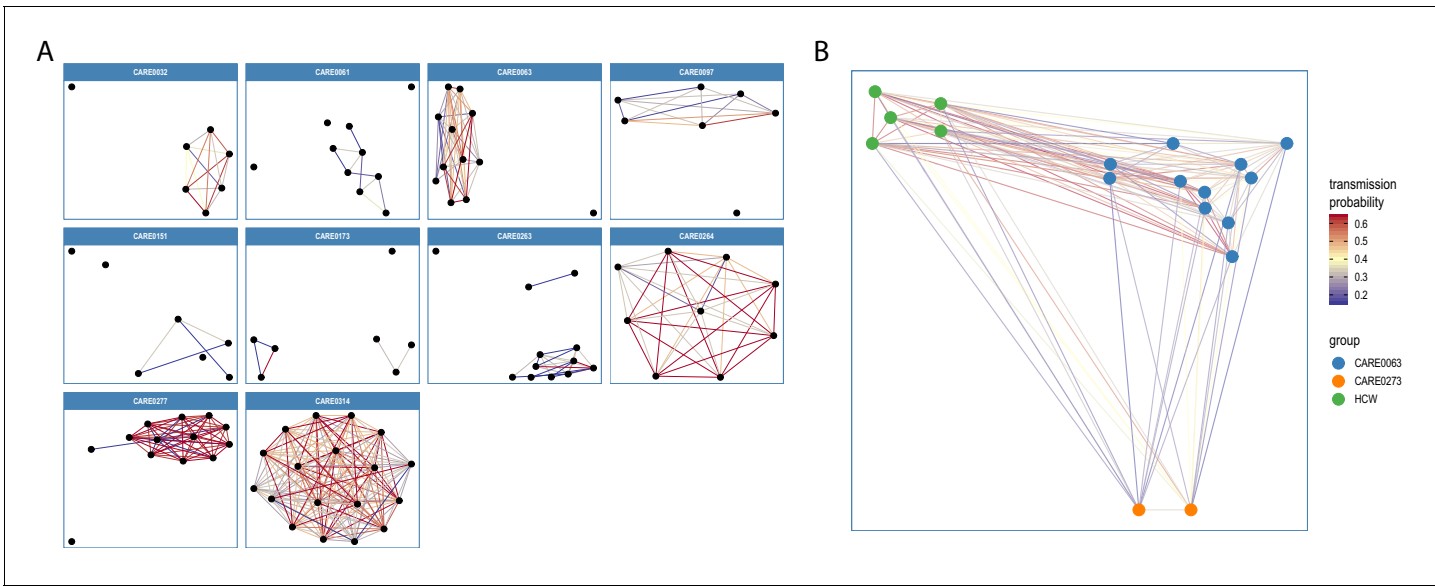

**Figure 7.** Visualisations of SARS-CoV-2 clusters among care home residents. Transmission networks were produced using a derivative of the *transcluster* algorithm, which incorporates pairwise date and genetic differences to estimate the probability of cases being connected within a defined number of intermediate hosts. Clusters were defined using a probability threshold of ≥15% for cases being linked by ≤2 intermediate hosts (further details in Materials and methods). (**A**) Transmission clusters for the ten care homes with the largest number of care home residents with available genomes. Consistent with ***Figure 6***, several of the 10 care homes with the largest number of genomes comprised single transmission clusters (e.g. CARE0314), while others contained two or more clusters consistent with multiple independent transmission sources among the residents. These data alone do not indicate where the residents acquired their infections, and hospital-acquired infections for some of the clusters is a possibility alongside multiple introductions into the same care homes. (**B**) Visualisation of transmission links between residents of two nearby carehomes and a group of healthcare workers (HCW). Two care homes, CARE0063 (blue) and CARE0273 (orange), each had strong transmission links identified with the *transcluster* algorithm to a group of HCW (green). The HCW comprised paramedics and care home carers – one working at CARE0063 and the other working at an unknown care home. We do not have confirmatory epidemiological data available, but this raises the possibility of the cases sharing a linked transmission network.

The online version of this article includes the following figure supplement(s) for figure 7:

**Figure supplement 1.** Transmission network diagrams for all care homes with two or more cases with genomic data.
**Figure supplement 2.** Histogram of pairwise transmission probabilities between care home samples.
**Figure supplement 3.** Transmission probability threshold vs number of care home clusters.
**Figure supplement 4.** Pairwise SNP difference distribution between samples within clusters.
**Figure supplement 5.** Pairwise date difference distribution between samples within clusters, aggregated across dataset.
**Figure supplement 6.** Distributions of date ranges (from first to last sampling dates) for care homes vs clusters.
**Figure supplement 7.** Pairwise date difference distribution between samples within each cluster.

**Table 6.** Outbreak characteristics for 10 care homes with the largest number of SARS-CoV-2 genomes.

Epidemiological characteristics of the 10 care homes with the largest number of genomes are shown. Collectively these comprised 102 cases (102/700 (14%) of the total number of care home cases with genomic data available). 'Cluster count' refers to the number of SARS-CoV-2 clusters within each care home defined by *transcluster* (described in Materials and methods and main text). 'Major cluster count' shows the count for the dominant cluster (with the largest number of cases) and its percentage contribution to total case numbers for each care home. 'Care home date range' indicates the number of days from first sample to last sample date for residents from each care home. 'Cluster date range' indicates the number of days from first sample to last sample date for residents from each cluster within that care home, as defined by the *transcluster* algorithm, also showing the sample count (n) for each cluster. Sampling dates used collection date if known, or receive date in the diagnostic laboratory if collection date was unknown. The date range for each care home is typically larger than the date range for clusters within care homes, except for single-cluster care homes like CARE0314. This is consistent with the *transcluster* algorithm defining groups of cases occurring closer together in time. While the care homes frequently had more than one introduction of the virus among residents (i.e. >1 clusters), there was usually a single dominant cluster responsible for the majority of cases. Individual counts of males and females for each care home are not shown as this generally gave counts of less than five, risking patient anonymity. Overall, there were 59/102 (57.8%) females for these 10 care homes.

| Care home code | Sample count | Age (median, IQR, range) | Ct values (median, IQR, range) | Cluster count | Major cluster count | Care home date range (days) | Cluster date range (days, sample count) |
|---|---|---|---|---|---|---|---|
| CARE0032 | 7 | 87 (IQR: 81–91, range: 56–93) | 23 (IQR: 22–24, range: 14–26) | 2 | 6/7 (85.7%) | 39 | 0 days, n = 1<br>10 days, n = 6 |
| CARE0061 | 10 | 88.5 (IQR: 87–92.2, range: 84–97) | 23 (IQR: 21.2–26.5, range: 12–33) | 4 | 7/10 (70%) | 38 | 0 days, n = 1<br>22 days, n = 7<br>0 days, n = 1<br>0 days, n = 1 |
| CARE0063 | 12 | 74.5 (IQR: 67.8–81, range: 42–94) | 23 (IQR: 20.8–27, range: 14–30) | 2 | 11/12 (91.7%) | 21 | 18 days, n = 11<br>0 days, n = 1 |
| CARE0097 | 7 | 90 (IQR: 82.5–92, range: 73–95) | 23 (IQR: 20.5–24, range: 17–27) | 2 | 6/7 (85.7%) | 28 | 0 days, n = 1<br>14 days, n = 6 |
| CARE0151 | 7 | 81 (IQR: 77–89, range: 69–96) | 20 (IQR: 19–25.5, range: 17–30) | 4 | 4/7 (57.1%) | 20 | 0 days, n = 1<br>0 days, n = 4<br>0 days, n = 1<br>0 days, n = 1 |
| CARE0173 | 7 | 81 (IQR: 77.5–94, range: 71–95) | 19 (IQR: 17.5–26, range: 15–27) | 3 | 3/7 (42.9%) | 21 | 0 days, n = 1<br>3 days, n = 3<br>0 days, n = 3 |
| CARE0263 | 12 | 85.5 (IQR: 81.8–90.5, range: 69–97) | 19.5 (IQR: 18.5–24.8, range: 14–29) | 3 | 9/12 (75%) | 3 | 3 days, n = 9<br>0 days, n = 2<br>0 days, n = 1 |
| CARE0264 | 9 | 91 (IQR: 82–95, range: 73–96) | 26 (IQR: 25–27, range: 18–29) | 1 | 9/9 (100%) | 14 | 14 days, n = 9 |
| CARE0277 | 13 | 84 (IQR: 82–89, range: 71–94) | 26 (IQR: 24–27, range: 23–29) | 2 | 12/13 (92.3%) | 13 | 13 days, n = 12<br>0 days, n = 1 |
| CARE0314 | 18 | 87.5 (IQR: 81.2–90.8, range: 74–97) | 24 (IQR: 22.2–26, range: 14–29) | 1 | 18/18 (100%) | 5 | 5 days, n = 18 |

30 days of diagnosis. Care home residents thus bore a high burden of COVID-19 infections and mortality.

A smaller proportion of care home residents were admitted to ICU compared with people who were not from care homes. What treatments a patient receives, including the invasive treatments provided in intensive care, are complex and individualised decisions based on risk-benefit assessments involving patients, their families and carers, and healthcare professionals (*ICS, 2020*; *NICE, 2020*). Of note, non-invasive respiratory support (such as continuous positive airway pressure, high-flow nasal oxygen therapy and non-invasive ventilation) are routinely provided outside ICU in many UK centres. Despite care home residents being at higher risk of severe COVID-19, and being under-represented in ICU, admission to ICU was still correlated with significantly increased mortality. This is likely because patients admitted to ICU have more severe disease, typically requiring more intensive treatments such as organ support.

**Table 7.** Hospitalisation data for the 700 care home residents with genomic data available 700/1167 (60.0%) care home residents identified in the study had genomic data available and were used to define care home SARS-CoV-2 clusters.

We investigated the proportions of these care home residents that were hospitalised and may have acquired their infections through interactions with hospitals. This was possible for 694/700 (99.1%) individuals who had NHS numbers documented that could be linked with national hospitalisation data. Being hospitalised due to COVOD-19 was defined as the date of first positive sampling being within 2 days prior to admission up to 7 days post-admission. Suspected hospital-acquired COVID-19 infections were defined as first positive test being 7 days or more after hospital admission date and prior to discharge date (N = 13) or within 7 days following hospital discharge (N = 27). Of the latter group, 10 individuals were admitted to hospital and discharged on the same day prior to their positive test, nine were admitted for 1–7 days, and eight had been admitted for greater than 7 days.

| Category | Counts (%) |
| --- | --- |
| Care home residents with genomic data | 700 |
| Care home residents with genomic data that could be linked to hospitalisation data | 694/700 (99.1%) |
| Hospitalised during study period | 470/694 (67.7%) |
| Hospitalised due to COVID-19 | 398/694 (57.3%) |
| Suspected hospital-acquired COVID-19 | 40/694 (5.76%) |
| Discharged within 7 days of positive test | 230/694 (33.1%) |

Viral clusters were defined within each care home by integrating temporal and genetic differences between cases. This provides a 'high resolution' picture of viral transmission; without genomic data, separate introductions of the virus occurring around the same time are impossible to distinguish. Care homes frequently experienced 'outbreaks' of multiple cases within clusters (the largest of which had >10 residents), consistent with substantial person-to-person transmission taking place within care homes. Care homes also frequently had multiple distinct clusters (up to 4), consistent with independent acquisitions of COVID-19 among residents – however, a single dominant cluster usually comprised the majority of samples within each care home. The majority of care home residents in the genomic analysis did not acquire COVID-19 in hospital. In the context of a national lockdown, the most likely location they acquired their infection was the care home. The high frequency of care home outbreaks may reflect the underlying vulnerability of this population to COVID-19 and the challenges of infection control in care homes. In contrast, the UK as a whole had an average of 2.37 people per household in *Office for National Statistics, 2019a* and in the East region only 2.2% of households were made up of two or more unrelated adults (6.2% in London) (*Office for National Statistics, 2019b*).

These findings emphasise the importance of limiting viral transmission within care homes in order to prevent outbreaks. Given there is increasing evidence for asymptomatic and presymptomatic transmission of SARS-CoV-2 (*Arons et al., 2020*; *Goldberg et al., 2021*; *He et al., 2020*), isolating residents or staff when they develop symptoms is not sufficient to prevent within-care home spread once the virus has entered the care home. Certain measures may be required on an ongoing basis within care homes when there is sustained community transmission, even when no outbreak is suspected (at least until the morbidity and mortality of the virus in older people has been reduced substantially through vaccination or treatments). These may include use of appropriate Personal Protective Equipment (PPE) for staff and visitors (including visiting healthcare professionals and friends and family), rigorous hand hygiene, social distancing, and making use of larger, well-ventilated rooms for social interactions or socialising outdoors, providing that this is practical and safe (*Jones et al., 2020b*). This is consistent with current national guidance for care homes in England (*Public Health England, 2020c*; *UK government, 2020b*). Face coverings for residents themselves when interacting socially in communal indoor areas could be considered, if acceptable to residents.

The majority of residents had hospital contact during the study period, indicating substantial opportunity for infections to pass between care homes and hospitals in either direction. A third of patients were discharged from hospital within 7 days of their first positive test, and thus were

potentially infectious at discharge. We identified transmission clusters that would be consistent with COVID-19 spread between care home residents and HCW, based both in the community and in hospitals. A previous study found that working across different homes was associated with higher SARS-CoV-2 positivity among staff (*Ladhani et al., 2020*). Limiting the spread of COVID-19 between care home residents, HCW and hospitals is a therefore another key target for infection control and prevention.

There are several limitations to this study. First, not all of the COVID-19 cases from the East of England have been included. Serology data suggest that 10.5% of all residents in care homes for people aged 65 and older in England had been infected with SARS-CoV-2 by early June, the majority of whom were asymptomatic (*UK government, 2020c*). The Cambridge CMPHL did not receive all the samples tested from the region; national data indicate around half of the COVID-19 cases reported from EoE during the study were included. Viral sequence data were not available for 40% of care home residents, as a result of missing samples, mismatches between sequences and metadata, genomes not passing quality control filtering using a stringent threshold (<10% missing calls), or sequences being unavailable at the time of data extraction. Viral cluster sizes may therefore be underestimated.

Second, the nature of diagnostic testing sites changed during the study period as regional hospitals developed their own in-house testing capacity and community testing laboratories were set up. 'Pillar 2' testing in the UK was outsourced to high-throughput laboratories during April 2020 and performed an increasing proportion of community testing. It is possible that some care home residents from the same care home could have been tested through different routes, with symptomatic cases more likely to be tested in 'Pillar 1' via the CMPHL (and included in this dataset), and asymptomatic screening occurring more via the Pillar two laboratories. However, most care homes in EoE only began systematic screening after the end of our study following the introduction of the UK care home testing portal on 11th May 2020. Moreover, the *transcluster* algorithm allows for 'missing links' within a cluster (the threshold used assumed a $\geq$ 15% probability of infections being linked within $\leq 2$ intermediate hosts), reducing the impact of missing care home cases on defined clusters. The changing profile of COVID-19 testing in the UK between March and May 2020 should therefore be factored into all interpretations of COVID-19 epidemiology from that period.

Third, defining who is a care home resident from large electronic healthcare records is challenging and, despite substantial efforts (described in Materials and methods), some care home residents may have been missed. Using pre-defined coding such as care home CQC registration numbers when patients are booked into hospital systems, rather than free-text data entry, would help considerably with care home surveillance. Multiple rounds of electronic searches and manual inspection were undertaken to identify as many care home residents as possible, and every care home resident included was cross-referenced against a CQC database of registered care homes in England. The care homes included for analysis should therefore be accurate.

Fourth, low viral sequence diversity limits the power of genomics to infer transmission clusters. Between-care home transmission was not investigated specifically because, unlike within-care home cases, opportunities for transfer of SARS-CoV-2 between care homes cannot be assumed or inferred from the data. This could be assessed in a dedicated prospective study gathering epidemiological data on between-care home contacts. Even within care homes, it is possible some genetically similar viruses are from unconnected introduction events. However, incorporating genomic data is more accurate for excluding linked transmission than if only temporal data are available. Genomics can thus be used to 'rule out' cases as being part of a linked cluster if the genetic difference is greater than would be expected given the viral mutation rate. This could be practically informative for care homes (along with other organisations at risk of COVID-19 outbreaks like factories *Middleton et al., 2020*), with implications for infection control procedures. Directionality of person-to-person transmission cannot be inferred from the *transcluster* algorithm. Inferring the likelihood of transmission direction between pairs of individuals requires integration with multiple forms of epidemiological data, yielding a probabilistic estimate (*Illingworth et al., 2020*).

In conclusion, care homes represent a major burden of COVID-19 morbidity and mortality, with transmission events introducing SARS-CoV-2 into care homes and subsequent transmission within them. Genomic data can be used in outbreak investigations to define viral clusters; this is critically dependent on integration with epidemiological data. The cut-offs we used for defining care home clusters were pragmatic but plausible given current understanding of the biology and epidemiology

of SARS-CoV-2. Such cut-offs can be helpful for producing understandable outputs for biological and public health interpretation (*MacFadden et al., 2018*; *Stimson et al., 2019*), and for focusing investigations with limited public health resources. Future work will need to prospectively integrate genomic and epidemiological data to rapidly identify viral clusters, thus enabling deployment of infection control and public health interventions in real time.

## Materials and methods

### Study overview

Data were collected on SARS-CoV-2-positive samples from the East of England, tested at the PHE CMPHL in Cambridge, between 26th February and 10th May 2020. The CMPHL is a PHE diagnostic laboratory that receives samples from across the East of England. The East of England is one of nine official regions in England. In the 2011 census, it had a population of 5,847,000, one of the fastest growing populations in England and Wales and the fourth largest population of the nine official regions (*Office for National Statistics, 2011*). The most populous cities include Luton, Norwich, Southend-on-Sea, and Peterborough (*City Population, 2020*). The 10th May was selected as a study end-date because it encompassed the bulk of the 'first wave' of the epidemic in the East of England. Furthermore, prior to the 11th May 2020, systematic screening of all residents within care homes was much less common and testing primarily occurred where there was a suspicion of an outbreak. The UK government launched a national care home testing portal on 11th May 2020 (*UK government, 2020d*), in which all care home staff and residents were eligible for testing with priority for homes caring for people aged 65 years or older. Ending the study on 10th May reduces the risk of bias which may be introduced by uneven systematic screening, for example when comparing the population genetics of care home and non-care home samples, if care homes undergo screening while non-care home settings do not. During the study period, the scope of testing in hospital, community, and care home settings changed several times, as eligibility criteria were modified (*Figure 1—figure supplement 1*). When interpreting trends in COVID-19 cases in the UK during this period it is essential to consider the changing capacity and policies surrounding testing.

### Diagnostic testing, metadata collection, and genome sequencing

For details on diagnostic testing, patient metadata collection, and nanopore genome sequencing see *Meredith et al., 2020*. Briefly, CMPHL used an in-house generated and validated one-step RT q-PCR assay detecting a 222 bp region of the RdRp genes, along with an MS2 bacteriophage internal extraction control, using the Rotorgene PCR instrument. Samples that generated a Ct value $\leq$36 were considered positive. The study aimed to sequence all samples which tested SARS-CoV-2 PCR positive at the CMPHL during the study period. Sequencing of every positive diagnostic sample could not be performed, however, for the following reasons: (i) sample unavailability (e.g. diagnostic samples being lost or discarded before they could be collected by the sequencing team); (ii) labelling errors when assigning sequencing codes (which resulted in specimens being discarded); or (iii) metadata mismatches (if the sample did not match to a metadata record downloaded from the hospital electronic patient records system). Samples were either sequenced on site using Oxford Nanopore Technologies or transported to the Wellcome Sanger Institute for Illumina sequencing.

Samples from Cambridge University Hospitals NHS Foundation Trust (CUH) and a selection of East of England (EoE) samples were sequenced on site to provide rapid information on hospital-acquired infections (*Meredith et al., 2020*). Nanopore sequencing (Oxford Nanopore Technologies) took place in the Division of Virology, Department of Pathology, University of Cambridge, following the ARTICnetwork V3 protocol and assembled using the ARTICnetwork assembly pipeline. The sequencing workflow involved a directional sample flow as used in a diagnostic laboratory which includes separated pre- and post-PCR areas, with dedicated equipment for each stage of the process. All steps were performed in PCR cabinets which were cleaned using DNA removal solutions and a UV decontamination cycle run after each batch. All sequencing batches included at least one water negative control carried over from the reverse-transcription step. Mapped reads were assessed in real-time during sequencing with RAMPART (*Hadfield, 2020*) and all data from batches containing a contaminated negative control were discarded before sequence assembly. The remaining EoE samples, where available, were sent to the Wellcome Sanger Institute (WSI) for sequencing.

Sequencing at WSI used Illumina technology. cDNA was generated from SARS-CoV-2 viral nucleic acid extracts and subsequently amplified to produce 400nt amplicons tiling the viral genome using V3 nCov-2019 primers (ARTIC). This was followed by Illumina library generation using the NEBNext Ultra II DNA Library Prep Kit for Illumina (New England Biolabs Inc, Cat. No. E7645L). Libraries were amplified with KAPA HiFi Ready Mix (Kapa Biosystems, Cat. No. 07958927001) and uniquely indexed with a 100 µM i5 and i7 primer mix (50 µM each) (Integrated DNA Technologies) to allow multiplexing of up to 384 SARS-CoV-2 viral extracts into one sequencing pool. The PCR products were pooled in equal volume and purified with an AMPure XP workflow (Beckman Coulter, Cat. No. A63880). The purified pool was quantified by qPCR (Illumina Library Quantitation Complete kit, Cat. No. KK4824) and sequenced on one lane of an Illumina NovaSeq SP flow cell (Illumina Inc, NovaSeq 6000 SP Reagent Kit v1.5 (500 cycles), Cat. No. 20028402), with XP workflow (Illumina Inc, NovaSeq XP two lane kit v1.5, Cat. No. 20043130). Genomes were generated for each library's sequencing data using bwa mem (*Li, 2013*) for alignment with MN908947.3 (*Wu et al., 2020*) as reference, samtools (*Li et al., 2009*) for pileup and ivar (*Grubaugh et al., 2019*) for trimming and consensus generation, all orchestrated by the ncov2019-artic-nf pipeline (*Bull, 2020*, cf01166, b88235d and 48816ee).

The WSI sequencing workflow also uses negative controls and the pass rate to date related to negative controls is 90%. Sequencing read counts are considered after a clipping and minimum alignment length filtering step (corresponding to data which is used to create consensus sequence or variant calls). Such read counts for the samples analysed in this study were typically in the millions (median: 4,497,543). If such read counts for the corresponding negative controls are >100 then the samples are currently failed. This QC procedure was introduced for samples analysed on or after the 18th of April. Of the 1007 samples analysed in this study sequenced at WSI (503 care home residents and 504 non-care home residents), 749 were sequenced once this workflow was established, 242 were sequenced before this but had a negative control and 16 did not have a negative control. If we apply the current criteria then 38 of these earlier samples would have failed (38/1400 = 2.7% of the analysed samples). Of these 38 samples, 26 are non-care home samples and 12 are from care homes. Of the 12 care home samples (12/700 = 1.7% total care home genomes analysed), one belongs to one of the 'top 10' care homes with the largest number of genomes, care home CARE0063, which comprises a single cluster of 12 genomes using the *transcluster* algorithm, described in main text. Thus, the main result of our genomic cluster analysis (that multiple introductions are often observed in care homes, but typically a single dominant cluster causes most of the cases) would not be altered by the small number of early genomes included that would now be excluded by current criteria.

Sequences were available from both Illumina and Nanopore platforms for eight care home residents included in the study (in all cases the Illumina data were used for the study analysis). In 7/8 cases, the sequence pairs were identical. In one case, there were two SNP differences between the consensus fasta sequences: C1884T and C16351T; for both SNPs, the Illumina sequence matched the reference genome (C) and the nanopore sequence had the alt call (T). These are not included among a list of previously identified sites that are highly homoplasic or have no phylogenetic signal and/or low prevalence (*De Maio and Walker, 2020*). The sequence pairs are shown below:

| Illumina sample - COG-UK ID | Illumina sample - date | Nanopore sample - COG-UK ID | Nanopore sample - date | Pairwise SNP difference |
|---|---|---|---|---|
| CAMB-761D5 | 30/03/2020 | CAMB-7B088 | 11/04/2020 | zero |
| CAMB-1AF1F0 | 30/04/2020 | CAMB-1AD8A2 | 30/04/2020 | zero |
| CAMB-1AE7C2 | 30/04/2020 | CAMB-1AC269 | 30/04/2020 | 2 |
| CAMB-80590 | 09/04/2020 | CAMB-789BD | 06/04/2020 | zero |
| CAMB-1AB23D | 20/04/2020 | CAMB-840B9 | 26/04/2020 | zero |
| CAMB-83AAD | 15/04/2020 | CAMB-8416B | 25/04/2020 | zero |
| CAMB-1ABE2A | 21/04/2020 | CAMB-8468A | 27/04/2020 | zero |
| CAMB-1AB631 | 21/04/2020 | CAMB-1ABF18 | 27/04/2020 | zero |

As with all the sample dates used, the above dates are based on sample collection date where available, with missing data substituted with the date of receipt in the laboratory. SNP differences were identified from a vcf file produced from the alignments using the package *snp-sites* v 2.5.1 (*Page et al., 2016*), command:

$$snp - sites - valignment\_file.aln$$

In *Meredith et al., 2020*, out of 14 sample pairs sequenced both by Illumina at WSI and nanopore in the University of Cambridge there were zero SNP differences at positions where both sequences had made a call (*Meredith et al., 2020*). There are several reasons why pairwise comparisons between different sequences from the same individual may not be identical, even if both sequences are produced using the same technology. When the cycle threshold (Ct) of a sample is near the limit of detection sensitivity, and/or RNA is degraded (e.g. due to delays between sampling and sequencing at room temperature), it is likely that amplicons that are not as efficiently amplified by the multiplex PCR may have low read coverage, or could be more sensitive to amplification bias. In this case, the samples both had high Ct values: CAMB-1AE7C2 (sequenced by Illumina at WSI) had Ct value of 30 and CAMB-1AC269 (nanopore sequenced in Cambridge) had a Ct value of 31. Median Ct value for the 700 care home residents with genomes analysed was 24 (interquartile range: 20–27) (data displayed in *Table 1*). If an individual is infected with more than one clone at significant frequency, it is also possible for stochastic variation in read counts for the two variants to yield different consensus calls at the variant locus. However, larger studies have systematically evaluated sequencing quality for SARS-CoV-2 between Oxford Nanopore Technology (ONT) and Illumina, and demonstrated highly accurate consensus-level sequence determination (*Bull et al., 2020*). Given this degree of consensus sequence accuracy, and because *transcluster* uses a transmission probability cut-off based on integrating pairwise SNP and temporal differences (rather than relying solely on a strict SNP cut-off), limited sequencing noise is unlikely to have a substantial impact on the clusters identified.

COG-UK IDs and GISAID accession numbers for genomes analysed in this study are included in Supplementary Materials, along with a complete author list for the COG-UK consortium.

## Sample selection

As described in *Meredith et al., 2020*, patient metadata were downloaded daily from the electronic medical record system (Epic Systems, Verona, WI, USA) and metadata manipulations were performed in R (v 3.6.2) using the *tidyverse* packages (v 1.3.0) installed on CUH computers. Positive samples were collected and assigned either for nanopore sequencing on site (focusing on CUH samples and a randomised selection of EoE samples), or sent to WSI for Illumina sequencing. Metadata were uploaded weekly to the MRC CLIMB system as part of the COG-UK Consortium. Samples included healthcare workers (HCW) tested in the CUH HCW screening programme (*Jones et al., 2020a*; *Rivett et al., 2020*), all of which were nanopore sequenced on site.

## Identifying care home residents

Care home residents were identified using a two-stage data mining approach followed by manual inspection and linking of putative care home addresses to care homes registered to the Care Quality Commission (CQC).

### Step 1: search terms in patient address fields

Patient address lines 1 and 2 were searched for the following list of key phrases (not case sensitive) in their electronic healthcare records; if any phrases were present the patient was labelled as being from a care home:

- residential home'
- care home'
- nursing home'
- care centre'
- care hom'
- nursing hom'
- residential hom'

- carehome'

This identified 765 patients as being care home residents.

## Step 2: matching location names to CQC registered care facilities

Many care homes do not have the above list of phrases in their address names. To capture these facilities, we used the publicly available database of care homes registered to the CQC, the independent regulator of health and adult social care in England. All organisations providing accommodation for persons who require nursing or personal care must be registered with the CQC, including care homes with or without nursing care (*Care Quality Commission, 2020b*). Details of the CQC registration scope can be found in 'The scope of registration (Registration under the Health and Social Care Act 2008)', March 2015, available at this link as of 24th June 2020: (*Care Quality Commission, 2015*).

The file 'CQC care directory – with filters (1 June 2020)' was accessed on 23rd June 2020 from the CQC website: (*Care Quality Commission, 2020c*), and the following filters were applied:

- Total facilities in CQC database: N = 49,516,516
- Carehome?' column filtered to 'Y': N = 15,507*
- Only care homes for which the 'Location Postal Code' column matched at least one postcode from the dataset of 6600 patients were included, yielding N = 444 care homes.**
- Following manual review and consistifying postcodes with the sample metadata, a set of 469 CQC registered care homes were included.***

*Filtering using the 'carehome?' column was based on advice given after correspondence with the CQC.

** Requiring CQC registered care homes to match postcodes from the patient dataset minimised the number of 'false positives' – patients whose address name matched a CQC registered care home name by coincidence.

*** 25 CQC registered care homes were added following manual review of the identified putative care home residents, who had a different postcode documented in the electronic healthcare records for the same care home, yielding the final 'CQC EoE care home search set' of 469 care homes.

We then used the values from the 'Location name' column of the filtered CQC dataset (i.e. the care home facility names) as search phrases for address line one in the patient database. Any patients with exactly matching phrases were labelled as care home residents. This increased the number of care home residents identified by a further 382–1147, that is, around one third of care home residents were identified using CQC facility names and would have been missed by relying on generic care home-related search phrases alone.

## Step 3: manual inspection and data clean up

Address lines for the non-care home patients were manually inspected; this identified a further 89 care home residents. Most of these had not been detected in steps 1 and 2 due to spelling or formatting issues with the patient addresses (e.g. short-hand abbreviations used for care home names, or inclusion of extra details like flat number meaning the string did not match a CQC care home name exactly).

Next, address lines for the care home residents were manually inspected and 14 were deemed not to be care home residents. Most of these were due to unrelated locations sharing the same address name as a CQC registered care home. The manual filtering steps thus yielded a care home resident count of 1147 + 89–14 = 1222. Address line 1 for all 1222 care home residents was manually inspected and formatted to ensure residents from the same care home had matching terms in this column. This was necessary due to discrepant address entrance formats for identical care homes; without this step, residents from the same care home would be incorrectly assigned to different anonymised care home codes.

## Step 4: linking care home addresses to CQC registered care homes

First line of patient address and postcodes were matched to care home names and postcodes from the CQC EoE care home search set (described above). Any discrepancies (care homes not matching the CQC data) were manually inspected and in the majority of cases the discrepancy could be

reconciled (e.g. alternative name or postcode used for the same care home). In 55 cases, a 'care home' was reclassified to non-care home, either because the address was independent housing with a matching name to a care home by coincidence, or because a care facility was determined by CQC definitions to not be a care home – for example several mental health community hospitals, drug rehabilitation centres, and supported living environments were excluded. This yielded the final analysis set of 1222–55 = 1167 care home residents, from 337 care homes. All 337 care homes included were therefore linked to CQC data; in two cases, the care home had been previously registered but had since been 'archived', and the most recent CQC data for defining whether residential or nursing care was being provided was used.

Care home location IDs assigned by the CQC were turned into anonymised codes (format: CARE followed by a four-digit numeric code). Care homes were classified as 'residential homes' or 'nursing homes' using the CQC data column 'Service type - Care home service with nursing' filtered to 'Y' for care homes with nursing, and column 'Service type - Care home service without nursing' = 'Y' for care homes without nursing ('residential homes'). If both fields were 'Y' then the care home was coded as being a nursing home.

## Linking care home data to CUH acute medical testing data

The dataset of 7407 PCR-positive samples with metadata were collected prospectively as part of the COG-UK study in Cambridge. Data on CUH acute care testing, including categorisations of whether infections were community- or hospital-acquired (definitions provided in *Meredith et al., 2020*) and data on patient outcomes (mortality at 30 days and ICU admissions), were collected separately as part of CUH and national monitoring. During the study period, 464 patients tested positive for COVID-19 at CUH.

When merging the metadata collected for COG-UK (including the above care home categorisations) with CUH acute testing data, 71 care home residents tested at CUH were identified. However, there were 23 samples that had tested positive in CUH that were not in the COG-UK dataset. Of 23, 21 of these were tested on the SAMBA platform at CUH (*Collier et al., 2020*), which is not PCR-based; sequencing was not possible for these samples owing to rapid RNA degradation. For technical reasons, SAMBA results were not included in the data collected prospectively in the Cambridge COG-UK study. The remaining two discrepancies were not captured in the electronic patient record downloads, which likely reflects periods where the download processes and coding methods were being established. Of the 23 missing samples, 20 were community-onset community-associated, two were hospital-onset indeterminate healthcare-associated, and one was a healthcare worker. These are counted as such and depicted with the above categorisations in the CUH epidemic curve shown in *Figure 3B*. Of the 23 CUH samples missing from the Cambridge COG-UK dataset, one was determined to be a care home resident, bringing the total CUH care home residents analysed to 72.

## Statistics

All statistical analyses were performed in R. The logistic regression model used to estimate odds of 30-day mortality was coded as follows: glm.fit <- glm(mortality_30_days ~ age + sex + care_status + ICU_admission + diagnostic_ct_value, data=data, family=binomial) summary(glm.fit).

Odds ratios and 95% confidence intervals were derived by exponentiating the model coefficients: exp(cbind(coef(glm.fit), confint(glm.fit))).

To produce the plot of odds ratios shown in *Figure 4*, the age and diagnostic Ct value continuous variables were transformed into binary categoricals using cut-offs of age $\geq$80 years and Ct value <20.

Wilcoxon rank sum tests performed in R using command format: wilcox.test(x, y, alternative = 'two.sided', conf.level = 0.95).

p-Values below $10^{-5}$ are not reported.

## Selecting randomised sample of non-care home residents as comparison group

A randomised sample of non-care home residents was selected to use as a control group for comparison of viral lineage composition against the care home residents. Because this group was

intended to be representative of non-care home community-acquired transmission, we applied the following inclusion criteria prior to randomisation:

- Patient address available.
- Not one of the identified care home residents.
- Not a healthcare worker (information only available for people tested at CUH).
- Not a CUH case of indeterminate, suspected or definite hospital acquired infection.
- Not living in a long-term care facility other than a care home (e.g. mental health hospital, rehabilitation unit, etc).
- Not living in a prison.

We attempted to have a roughly equivalent representation of nanopore and WSI sequenced samples as present in the care home database. Samples were selected using the R randomisation command *sample_n()* from available genomes in the CLIMB database passing QC filters. Having identified 698 samples, any cases with matching addresses that had been excluded were added to yield the final set of 700 non-care home genomes for comparison. Of the 700 non-care home samples included, we note that there were five instances of pairs of samples sharing the same address; in all five cases the pairwise SNP difference was zero or 1, and in 4/5 cases the people shared the same surname. This non-care home comparison set is not part of the care home viral cluster analysis performed using the *transcluster* algorithm.

## Care home viral phylogenetics and cluster analysis

Consensus fasta sequences were downloaded from the MRC-CLIMB website (https://www.climb.ac.uk/) (*Connor et al., 2016*). Genomes were de-duplicated (one genome per person) and passed through quality control (QC) filtering using the same criteria as in *Meredith et al., 2020*: genome size >29 Kb, N count <2990 (i.e. >90% coverage). Where there were multiple sequences from the same patient, the sequence passing QC filters that was collected first was used for genomic analysis (closest to the onset of symptoms).

The 700 de-duplicated viral genomes from care home residents passing QC were aligned using MAFFT (v 7.458) (*Katoh and Standley, 2013*) with default settings. Command: '/PATH/mafft' −retree 2 −inputorder 'multi_fasta_filename.fasta' > 'alignment_filename'.

A SNP difference matrix was produced from the alignment using *snp-dists* v 0.7.0 (*Seemann, 2020*) installed in a conda environment, run with the following command: snp-dists -c alignment_filename.aln > snp_diff_matrix_filename.csv.

The SNP difference matrix was manipulated in R using the *Matrix* and *tidyverse* packages to generate the SNP difference histogram and boxplots.

Phylogenetic trees were generated using IQ-TREE (v 1.6.12 built 15th August 2019). An alignment was generated as above including a reference genome from Wuhan, China, collected December 2019 and used to root the tree (GISAID ID: EPI_ISL_402123). The IQ-TREE Model Finder Plus option was used (*Kalyaanamoorthy et al., 2017*) which searches from a database of available nucleotide substitution models and selects the best fit to the analysis, command line:

$$\sim/PATH/iqtree - s\,alignment\_filename - m\,MFP$$

The best-fit nucleotide substitution model according to BIC was GTR+F+R2. The tree shown in this manuscript was produced using the GTR+F+R2 model with the ultrafast bootstrap option (*Hoang et al., 2018*) run through 1000 iterations to estimate branch support values, using command:

$$\sim/PATH/iqtree - s\,alignment\_filename - m\,GTR + F + R2 - bb\,1000$$

Newick trees were manipulated in *FigTree* (v 1.4.4) to root on the Wuhan sample and put in increasing node order. Trees were visualised initially using the microreact online tool (*Argimón et al., 2016*), and *Figure 6A* was produced in R using *ggtree* (v 2.0.4) (*Yu et al., 2017*).

For the phylogenetic tree of all samples in the study (*Figure 6—figure supplement 1*), consensus fasta files were downloaded from the COG-UK database (https://www.cogconsortium.uk/data/) accessed 01/12/2020. The same QC filtering described above was applied (genome size >29 Kb, N count <2990). Sequences passing QC were linked by their COG-UK IDs to individuals from this study. Of the 6600 people in the study, 1167 had been identified as care home residents and 700/

1,167 (60.0%) had genomes available that passed QC at time of the main analysis, leaving 5246 non-care home residents (187 were undetermined). Of the 5246, 3745 (71.4%) non-care home residents had genomes available that passed QC (including the 700 randomly sub-sampled non-care home residents described above). A multiple sequence alignment was produced in MAFFT and phylogenetic tree produced using IQTREE, command line:

$$iqtree - s\,alignment\_all.aln - mGTR + F - ntAUTO - ntmax16 - mem16G - bb1000$$

The tree was manipulated in *FigTree* (v 1.4.4) and *Figure 6—figure supplement 1* was produced in R using the *ggtree* package as with *Figure 6*.

## Lineage assignment

Viral lineages were assigned using the Pangolin COVID-19 Lineage Assigner web utility (*COG-UK, 2020*). Analysis was performed with Pangolin (*Rambaut et al., 2020a*) version 1.1.14, lineages version 2020-05-19-2. Contextual information about lineages was taken from *Rambaut et al., 2020b*, accessed 24/07/2020.

## Clustering

Clusters were produced using an implementation of the *transcluster* algorithm (*Stimson et al., 2019*; *Tonkin-Hill, 2020*). Instead of targeting the number of SNPs separating two genomes, the *transcluster* algorithm proposes a probabilistic alternative which estimates the number of intermediate transmission events separating two sampled genomes. The method takes into account both genetic SNP distance as well as the time at which each sample was taken. The approach models both the SNP distance and the number of intermediate hosts as a Poisson process. Using a predefined evolutionary rate as well as an estimate of the generation time (the time between transmission events), the method infers the distribution of the number of intermediate hosts separating two samples.

Briefly, $N$ let be the SNP distance separating two genomes and $\delta$ the time difference between when the samples were taken. We would like to estimate $h$, the time between the infection times of the two samples. The number of SNPs per unit time can be modelled as a Poisson process with evolutionary rate $\lambda$. Similarly, we assume the rate $\beta$ at which the pathogen jumps to a new host is constant resulting in another Poisson process for the number of intermediate hosts given $h$ and $\delta$. We are thus interested in the probability that there are $\kappa$ intermediate hosts given $N$ and $\delta$ which, following the derivation in *Stimson et al., 2019*, can be written as:

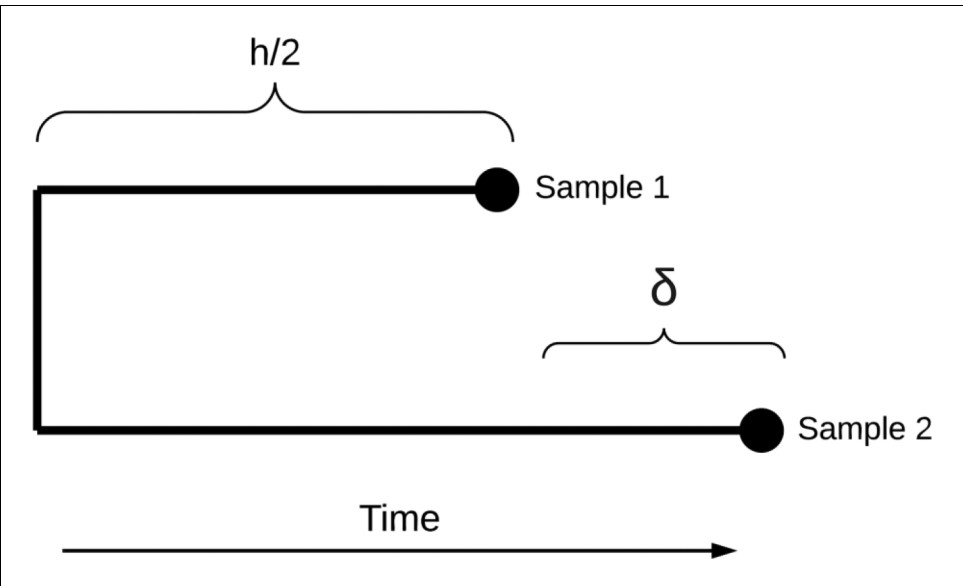

**Scheme 1.** Diagram representing transmission dynamics between two samples.

$$P(k|N,\delta) = \int_{h=0}^{\infty} \mathcal{L}(h|N,\delta)P(k|h)dh$$

This can be expressed as the sum:

$$P(k|N,\delta) = \frac{\lambda^{N+1}\beta^k(n+k)!}{e^{\delta\beta}n!k!\sum_{i=0}^{N}\frac{(\lambda\delta)^i}{i!}}\sum_{i=0}^{N+k}\frac{\delta^{N+k-i}}{(N+k-i)!(\lambda+\beta)^{i+1}}$$

The implementation of *transcluster* assumed a viral mutation rate of 1e-3 substitutions/site/year (*Fauver et al., 2020*) and generation time of 5 days, approximated by previous estimates of the serial interval of SARS-CoV-2 (*He et al., 2020*; *Zhang et al., 2020*). Days between first positive sampling date for pairs of individuals was used as a proxy for generation time. As above, where collection date was missing, the date the sample was received in the Cambridge PHE laboratory was used. The resulting pairwise transmission probabilities were used to generate a pairwise distance matrix and clustering was performed using single linkage hierarchical clustering with the R *hclust* function. Links were only considered if they involved residents from the same care home; thus, the largest theoretical number of clusters in this analysis would be 700 (every individual is their own distinct cluster), and the smallest would be 292 (one cluster for each care home).

The relationship between the probability of infections being linked by ≤2 intermediate hosts and the resulting number of care home clusters was explored. A higher threshold leads to more care home clusters, with greater likelihood of linked transmission within each cluster than when using a lower threshold. A pragmatic cut-off of ≤15% probability was selected, yielding 409 clusters. The majority of pairwise comparisons within clusters were zero or 1 SNP different and <14 days apart.

For 16/700 (2.3%) genomes, the sample that produced the analysed sequence was not the first positive test for that individual in the dataset. This could have occurred if the first positive test was not sequenced, or the sequencing failed or did not pass QC filters. This could theoretically lead to different clustering outcomes, if two cases were counted as further apart temporally than they really were from the date of first positive swab. To ensure this had not biased our findings, the *transcluster* analysis was re-run with identical thresholds using the date of first positive test for each individual (keeping the same genomes). There was no change in the number of clusters identified (n = 409).

To maintain study participant anonymity, care home residency status cannot be released publicly linked to their COG-UK genome codes. However, an anonymised version of the same dataset analysed in this study, with COG-UK sequence codes replaced by anonymised sample codes, can be accessed via GitHub at https://github.com/gtonkinhill/SC2-care-homes-anonymised. This includes all code and anonymised input data to reproduce the transmission analysis. Further discussion on data release is provided in Supplementary Materials.

## Investigating hospital admissions for care home residents

Hospital Episode Statistics (HES) data from 26th February to 10th May 2020 were linked to cases from this study using matching NHS numbers. The data were accessed by the Public Health England Healthcare Associated Infections (HCAI) division via the PHE Data Lake. This was possible for 694/700 (99%) of the care home residents with genomes available (used in the cluster analysis); six cases could not be linked to admission data due to missing NHS numbers in the study metadata.

Hospital admission coding included transfer of care between medical units as separate admissions. These were condensed into single admissions if the time interval between the preceding discharge and the following admission was less than or equal to 1 day; that is an admission had to occur 2 days or more after the preceding discharge to be counted as a new admission.

## Hospital admission data were parsed to yield the following outputs

- COVID-19-related hospital admission: first positive test date was −2 to +7 days inclusive from a hospital admission date
- Suspected hospital acquired: first positive test date was +7 days from a hospital admission to +7 days from a hospital discharge, inclusive. The people testing positive in the community within 7 days of discharge from hospital are categorised as, 'community onset, suspected

hospital acquired'; the people testing positive after 7 days from admission but before their discharge are categorised as, 'hospital onset, suspected hospital acquired'.
- For the six individuals with no NHS number, we assumed they were not discharged within 7 days of a positive test.

For the care home residents with community-onset, suspected hospital-acquired infections, the number of days the patient had been admitted to hospital prior to their positive test was calculated.

### CUH HCW-care home resident cluster analysis

The analysis of transmission between healthcare workers (HCW) and care home residents focused on CUH cases, where the richest metadata was available including HCW status.

Of 6600 PCR-positive patients, 91 had been identified as HCW. Of these, 74 were from the CUH HCW screening programme (which includes symptomatic, asymptomatic and household contact arms) (*Jones et al., 2020a*; *Rivett et al., 2020*) and 17 had presented acutely to CUH medical services, and been identified as HCW during their initial medical clerking and subsequent note reviews. Of the 91 HCW, 76 had genomes available for analysis (breakdown: 56 samples identified through the CUH HCW screening programme, 9 CUH HCW who presented to acute medical services at CUH, and 11 HCW from community settings (paramedics and care home workers) that had been flagged as HCW through admission clerkings). Of 464 CUH cases in the study period, 72 were care home residents (described above) and 54 of these had available genomes for analysis. The total combined analysis set of CUH HCW and care home residents was therefore 76+54 = 130.

The 130 genomes were aligned using MAFFT and underwent the same cluster analysis using the *transcluster* algorithm as described above. Transmission links between care homes were excluded as were links between HCWs. HCWs could belong to multiple clusters from different care homes to allow for the possibility of a HCW seeding multiple care home infections. Twenty-one clusters involving both care home residents and HCWs were identified. Of the 54 care home residents, 38 had links with HCWs within the 0.15 probability threshold. Medical notes for potential care home resident-HCW transmission pairs were reviewed by author WLH as described in *Meredith et al., 2020*, with cases being categorised as strongly linked epidemiologically (e.g. the HCW documented in the care home residents' medical notes); possibly linked (e.g. both working in the hospital at the same time but not in the same wards); or no evidence of an epidemiological link.

## Acknowledgements

We gratefully acknowledge the invaluable contributions of all members of the Wellcome Sanger Institute Covid-19 Surveillance Team (www.sanger.ac.uk/covid-team) who have supported this project. We would also like to thank Nick Donnelly for advice with statistical analyses, and the Public Health England Hospital Acquired Infection (HCAI) division, in particular Rebecca Guy and Mehdi Minaji, for assistance accessing hospital admission data for this study.

## Additional information

### Competing interests

M Estee Torok: I have received grant support from the Academy of Medical Sciences, the Health Foundation, and the NIHR Biomedical Research Centre. I have also received book royalties from Oxford University Press and honoraria from the Wellcome Sanger Institute. The other authors declare that no competing interests exist.

### Funding

| Funder | Grant reference number | Author |
| --- | --- | --- |
| Medical Research Council | COG-UK MC-PC-19027 | Sharon J Peacock |
| National Institute for Health Research | COG-UK MC-PC-19027 | Sharon J Peacock |
| Wellcome Trust | COG-UK MC-PC-19027 | Sharon J Peacock |

| | | |
|---|---|---|
| Wellcome Trust | Senior Fellowship 207498/Z/17/Z | Ian G Goodfellow |
| Academy of Medical Sciences | Clinician Scientist Fellowship | M Estee Torok |
| Health Foundation | Clinician Scientist Fellowship | M Estee Torok |
| National Institute for Health Research | | William L Hamilton Emily R Smith Ben Warne M Estee Torok |
| Wellcome Trust | Collabrative grant 204870/Z/16/Z | Charlotte J Houldcroft |

The funders had no role in study design, data collection and interpretation, or the decision to submit the work for publication.

### Author contributions

William L Hamilton, Conceptualization, Resources, Data curation, Formal analysis, Supervision, Visualization, Methodology, Writing - original draft, Project administration; Gerry Tonkin-Hill, Conceptualization, Data curation, Software, Formal analysis, Methodology, Writing - review and editing; Emily R Smith, Dinesh Aggarwal, Ben Warne, Colin S Brown, Resources, Data curation, Formal analysis, Writing - review and editing; Charlotte J Houldcroft, Data curation, Formal analysis, Writing - review and editing; Luke W Meredith, Resources, Data curation, Formal analysis, Supervision, Investigation, Writing - review and editing; Myra Hosmillo, Aminu S Jahun, Investigation, Writing - review and editing; Martin D Curran, Surendra Parmar, Sarah L Caddy, Anna Yakovleva, Theresa Feltwell, Malte L Pinckert, Yasmin Chaudhry, Sonia Gonçalves, Roberto Amato, Mathew A Beale, Michael Spencer Chapman, David K Jackson, Ian Johnston, Alex Alderton, John Sillitoe, Cordelia Langford, Resources, Investigation, Writing - review and editing; Laura G Caller, Fahad A Khokhar, Iliana Georgana, Resources, Investigation; Grant Hall, Resources, Data curation, Investigation, Writing - review and editing; Ewan M Harrison, Resources, Project administration, Writing - review and editing; Nicholas M Brown, Resources, Supervision, Investigation, Writing - review and editing; Gordon Dougan, Supervision, Project administration; Sharon J Peacock, Supervision, Funding acquisition, Project administration, Writing - review and editing; Dominic P Kwiatowski, Resources, Supervision, Project administration, Writing - review and editing; Ian G Goodfellow, Resources, Data curation, Supervision, Investigation, Project administration, Writing - review and editing; M Estee Torok, Conceptualization, Resources, Data curation, Formal analysis, Supervision, Investigation, Methodology, Writing - original draft, Project administration, Writing - review and editing; COVID-19 Genomics Consortium UK, Software

### Author ORCIDs

William L Hamilton https://orcid.org/0000-0002-3330-353X
Gerry Tonkin-Hill https://orcid.org/0000-0003-4397-2224
Dinesh Aggarwal http://orcid.org/0000-0002-5938-8172
Charlotte J Houldcroft http://orcid.org/0000-0002-1833-5285
Myra Hosmillo http://orcid.org/0000-0002-3514-7681
Aminu S Jahun http://orcid.org/0000-0002-4585-1701
Sarah L Caddy http://orcid.org/0000-0002-9790-7420
Grant Hall http://orcid.org/0000-0003-3928-3979
Iliana Georgana http://orcid.org/0000-0002-8976-1177
Nicholas M Brown http://orcid.org/0000-0002-6657-300X
Mathew A Beale http://orcid.org/0000-0002-4740-3187
Michael Spencer Chapman http://orcid.org/0000-0002-5320-8193
David K Jackson http://orcid.org/0000-0002-8090-9462
Sharon J Peacock http://orcid.org/0000-0002-1718-2782
Ian G Goodfellow https://orcid.org/0000-0002-9483-510X
M Estee Torok https://orcid.org/0000-0001-9098-8590

## Ethics

Human subjects: This study was conducted as part of surveillance for COVID-19 infections under the auspices of Section 251 of the NHS Act 2006. It therefore did not require individual patient consent or ethical approval. The COG-UK study protocol was approved by the Public Health England Research Ethics Governance Group (reference: R&D NR0195).

## Decision letter and Author response

Decision letter https://doi.org/10.7554/eLife.64618.sa1
Author response https://doi.org/10.7554/eLife.64618.sa2

# Additional files

## Supplementary files

• Supplementary file 1. Supplementary materials for 'Genomic epidemiology of COVID-19 in care homes in the East of England'.

• Transparent reporting form

## Data availability

The main analysis set comprised 700 genomes from care home residents. Additionally, a randomised selection of 700 genomes from non-care home residents was used for comparing lineage composition, and genomes from 76 healthcare workers tested at CUH were included for the analysis of care home resident-HCW transmission. Consensus fasta sequences for the 1,476 genomes are publicly accessible through the COG-UK website data section (https://www.cogconsortium.uk/data/). COG-UK also regularly deposits data into public databases such as GISAID (https://www.gisaid.org/). COG-UK sequence codes, GISAID accession IDs and virus names for the 1,476 analysed genomes are included in Supplementary file 1. Sequences generated through the COG-UK consortium have associated public metadata (available via the COG-UK website or GISAID), including patient age, sex, collection date (if available), and location to the level of UK county. COG-UK samples are sequenced under statutory powers granted to the UK Public Health Agencies. Matched patient data is securely released to the COG-UK consortium under a data sharing framework which strictly controls the handling of patient data. The status of individuals living in a care home and groups of such care home patients are both on the consortium restricted data list. This means that this data cannot be publicly released linked to their sequencing identifiers (eg. COG-UK sequence codes). This is because of the risk of deductive disclosure, potentially compromising study participant anonymity. However, code to fully reproduce the transcluster transmission analysis using anonymised metadata is available via GitHub at: https://github.com/gtonkinhill/SC2-care-homes-anonymised (v0.1.0). The genomes are the same as those used in the study, but sample names in the genetic distance matrix and corresponding metadata have been changed from COG-UK sequence codes to anonymised sample codes. The metadata (sampling dates) has been altered from the original patient data but in a way that preserves the date-differences between samples within care homes, thus yielding an identical transcluster analysis. If a researcher requires access to restricted metadata (including care home residency status) linked to the COG-UK sequence codes, then this will require a formal data sharing agreement with the COG-UK Consortium. Access to patient outcome information for patients treated at Cambridge University Hospitals NHS Foundation Trust (CUH) requires a data sharing agreement with CUH. Data will only be shared for public health and research purposes, not for commercial enterprise, and only to individuals working at reputable research and public health institutions for which data security can be assured. Should this be required researchers should contact the study corresponding authors in the first instance.

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
