## [Decision Letter]

**Acceptance summary:**

Given the importance of understanding the role of care homes and burden of SARS-CoV-2 in older individuals, understanding how transmission is occurring within and between these facilities is of key epidemiological importance. Further integrating genomic with epidemiological information can provide novel insight into transmission patterns that would otherwise be nearly to disentangle. This manuscript is able to leverage a wealth of information to add important insight into transmission cluster that can illuminate important factors dictating the epidemiological dynamics.

**Decision letter after peer review:**

Thank you for submitting your article "Genomic epidemiology of COVID-19 in care homes in the East of England" for consideration by *eLife*. Your article has been reviewed by two peer reviewers, and the evaluation has been overseen by a Reviewing Editor and Miles Davenport as the Senior Editor. The reviewers have opted to remain anonymous.

The reviewers have discussed the reviews with one another and the Reviewing Editor has drafted this decision to help you prepare a revised submission.

Additional background needed: transcluster method, Meredith paper, Figure 6.

Sequencing discrepancy:

This manuscript combines sequencing and epidemiological analysis to understand SARS-CoV-2 transmission in care home settings. The authors aim to answer questions related to the burden of care home-associated cases, outcomes for care home residents, and the transmission dynamics of the virus within care homes. These questions are laid out very clearly in this well-written manuscript, and the results are thoughtful and informative. The authors also do a nice job of outlining the limitations of their method in the Discussion.

1) While Figure 5 compares the viral lineages observed in care and non-care home individuals, I did not see non-care home sequences on the phylogenetic tree in Figure 6. Why is this? Incorporating additional sequences from individuals not in care homes from the same region (from both this study and other previously published studies, if available) could reinforce that there were transmission clusters within care homes (i.e., if all care home sequences were more closely related to each other than any other sequences). It appears that there are sequences from both and showing the relationship between care homes and non-care homes would help make sense of their results and ensure that the clustering conclusions are not artifacts of the sampling method. Why were only a few of the 10 care homes further discussed in the paper. Also, minor point but panel A is not further discussed in the manuscript.

2) Did the sequencing include the use of negative controls to detect possible contamination? Contamination is a common issue when using the ARTIC primers due to the many cycles of PCR amplification, and I have a hard time accepting sequencing data produced using this method without explicit description of the contamination controls or clean lab practices used. This information would be very helpful to see in the Materials and methods and would help promote careful validation of sequencing data.

3) Similarly, did the authors investigate the ONT/Illumina sequence pair that had two SNPs between the sequences? There is evidence that ONT sequencing in particular has biases at particular areas within the SARS-CoV-2 genome (e.g., https://virological.org/t/issues-with-sars-cov-2-sequencing-data/473, https://www.medrxiv.org/content/10.1101/2020.08.13.20174136v2), and it would be interesting to know if this discrepancy occurs at one of these regions, which would allow the authors to explain and possibly even correct the error. If the error cannot be easily explained, the authors should comment on why they don't think their analysis, which is based heavily on single-SNP differences between sequences, could be affected by whatever might be causing this discrepancy.

4) Overall the manuscript is fairly dense with a substantial amount of results and information provided in the text which would be better summarized as tables. For example, the authors state that 7,406 samples from 6,600 individuals were identified, but only 18% are from care homes? If so, is the 18% that was primarily analyzed in the manuscript? Or is this 18% only residents (and if so, are the care givers omitted?). Further, there are many values without percentages presented and it is difficult to follow which samples were considered in each section. For example, in the next section they report that 7% are from a hospital, but are these a subset of the first section? In general, it is difficult to follow how the samples/information in each section relate (or do not relate) to subsequent sections and exactly which samples (including the numbers and percentages) were analyzed.

5) The cluster results should provide more background and clarification. For example, it appears that only 60% of the data was analyzed in this section however it is not clear how this was decided. The authors should provide more information about the transcluster method since it is a newer method. This can help put the results in context, because currently is it not clear if the analysis considers all of the clusters together or only within cluster transmission. In addition, the authors state that 8 SNPs separate each care home, however they do not provide any indication if this suggest very close transmission amongst care homes. Additional lineage information (including statistics) would be helpful to put the data into context. Further, for outbreaks identified, information should be provided on the time these samples were collected.

Further, the sentence "We investigated the role of genomics in defining care home clusters by repeating the transcluster algorithm using the same parameters as for the main analysis but assuming all genomes were identical " is not clear enough. What authors mean by "the role of genomics?". Finally, a figure/table with SNPs, days separating and number of individuals per cluster would helpful since there is currently limited investigation provided. There is limited depth provided when exploring these clusters – for example the authors discuss a paramedic in cluster CARE0063 but they do not provide information of relatedness (SNP distance), how this sample related to the other samples in time, or any other information. In addition, if directionality cannot be inferred, this should be addressed within the limitations in the Discussion.

[Editors' note: further revisions were suggested prior to acceptance, as described below.]

Thank you for submitting your article "Genomic epidemiology of COVID-19 in care homes in the East of England" for consideration by *eLife*. Your article has been evaluated by the previous reviewers, and the evaluation has been overseen by a Reviewing Editor and Miles Davenport as the Senior Editor.

Summary:

Based on the revision, we believe there are still a few areas that require revision and additional detail prior to acceptance. We recognize the importance of the ethical and IRB process, however it is not clear why some information about the date samples were collected could not be provided. Oftentimes IRBs will allow the time between samples or more aggregated dates to be provided. Given that the method used is highly reliant on the timing of the samples, this information is needed to evaluate the work.

Essential Revisions:

1) Timing of samples: Information is needed to put the timing of samples into context – given that the method is highly reliant on this information. For example, were distinct samples taken a substantial period of time apart could possibly be part of the same transmission chain if not all cases were sampled? For example, dates by epi week or month/year are often allowed by IRBs. Or perhaps the authors could add some summary statistics to Table 6, e.g. span of time between samples in the care home and mean/spread of days between samples? This would be aggregate and could not be connected to a specific sample. Without additional information, the work is not transparent to the level that it can be properly evaluated. Additionally, we may have missed it but we don't think the authors specified the generation time they are using when running transcluster? Details on this and the uncertainty around this estimate would help, especially when compared to even relative/aggregate timing information (and a few more details on sampling). For example, we would be quite convinced that two samples taken 1-2 months apart could be ruled out to be separate introductions if there were no confirmed cases between them and the generation time was a few days with a small confidence interval.

2) Sampling bias: The revision does not adequate address the issues of sampling bias. For example, what was the depth of sampling in each care home? Were all positive cases sequenced? Only a proportion? What sampling bias (and subsequent impact on the results) may arise from the sampling strategies used? For example, the text states that: there is genomic information for "for 700 / 1,167 (60.0%) care home residents from 292 care homes" but only only 10 of these care homes were analyzed (102/700 total samples). It is not clear why all of the samples were not included and should be justified or these additional samples should be included in the analysis. This was included in the original reviews, however was not adequately addressed in the revision.

3) Provide additional detail on the phylogeny: Additional information is needed in the text to ensure that the conclusions based on SNP differences are supported by the phylogeny. For example, seeing the tree Figure 6A in more detail (e.g. a zoom in panel), would allow the reader to see if non-care home sequences fall within the colored clusters or not. This would also help address the issues of sampling bias (care and non-care home individuals) and the subsequent implications. For example, the authors say "Samples from the ten care homes with the 1180 largest number of genomes are highlighted by coloured circles on branch tips." : if I look at the first colored circle in cyan corresponding to CARE0151, based on Table 6 I would expect to see 7 samples, but I only count 5. It is overall difficult to deciphering the tree. The addition of more information in the figure legend would also be beneficial and avoid the reader from searching the information in the text. Nonetheless, it is surprising that the transcluster method, that defines clusters based in genetic information but also date of collection, identifies clusters with sequences that are scattered all over the tree like in the case of clusters CARE0151, CARE0277, CARE0061 or CARE0032. The fact that the clusters are heterogeneous is reflected in the pairwise SNP difference plot, and very clearly for CARE0277 that seems to have two sub-populations, but does not appear to be the case for CARE0151. These might need more in depth explanation despite the observation reported in “By contrast, several care homes were “polyphyletic”, with cases distributed across the phylogenetic tree and higher pairwise SNP difference counts between samples, consistent with multiple independent introductions of the virus among residents.”, for example.

[Editors' note: further revisions were suggested prior to acceptance, as described below.]

Thank you for resubmitting your work entitled "Genomic epidemiology of COVID-19 in care homes in the East of England" for further consideration by *eLife*. Your revised article has been evaluated by Miles Davenport (Senior Editor) and a Reviewing Editor.

Unfortunately the Editors and Reviewers felt that the manuscript is not acceptable for *eLife* as it has not reached the level of clarity needed to allow for the key conclusions to be evaluated and for someone else to replicate these results. In particular, the key piece of information that is missing is the amount of time between samples and clusters within the same care home that are needed to understand how the genetic data is being used to determine the clusters. While Table 6 is helpful, it still does not provide enough detail about the timing of infections – in particular the time between clusters within a care home and the mean time/distribution of time between samples within and between the care homes. These data are needed to interpret the results. Given limitations to protect the privacy of participants, information by care home – including clusters identified, dates associated with these clusters, and sampling timeframes – is acceptable as an alternative to including the care home information per sample.

Further, since the authors make a point to distinguish between monophyletic and polyphyletic cluster, a full tree should be included in the supplementary information (while the added zoom is a nice addition, it is lacking since it is just about one cluster).

Finally, the data sharing agreement does not meet the standards required for *eLife*. That is, simply stating that others may "discuss the process of signing a data sharing agreement" appears particularly subjective. The authors need to have a full data package available to anyone who requests it – or any limitations on providing the full data need to be specified. Eg: if it is limited to academic (non-commercial) study / people in a particular geographical jurisdiction / a confidentiality agreement is required – a draft of this agreement should be provided and a clear statement on what reasons for declining would be. These raw data need to be accessible to allow attempts at replication as an important part of the scientific process.

---

## [Author Response]

Additional background needed: transcluster method, Meredith paper, Figure 6

We have provided further background on the *transcluster* method and the Meredith *et al.* paper, provided further detail on viral clustering in care homes, produced a new version of Figure 6 allowing for additional conclusions, and added tables to present more of the data succinctly, as described further below.

Sequencing discrepancy:This manuscript combines sequencing and epidemiological analysis to understand SARS-CoV-2 transmission in care home settings. The authors aim to answer questions related to the burden of care home-associated cases, outcomes for care home residents, and the transmission dynamics of the virus within care homes. These questions are laid out very clearly in this well-written manuscript, and the results are thoughtful and informative. The authors also do a nice job of outlining the limitations of their method in the Discussion.1) While Figure 5 compares the viral lineages observed in care and non-care home individuals, I did not see non-care home sequences on the phylogenetic tree in Figure 6. Why is this? Incorporating additional sequences from individuals not in care homes from the same region (from both this study and other previously published studies, if available) could reinforce that there were transmission clusters within care homes (i.e., if all care home sequences were more closely related to each other than any other sequences). It appears that there are sequences from both and showing the relationship between care homes and non-care homes would help make sense of their results and ensure that the clustering conclusions are not artifacts of the sampling method. Why were only a few of the 10 care homes further discussed in the paper. Also, minor point but panel A is not further discussed in the manuscript.

We have produced a new version of Figure 6, which includes the 700 care home resident genomes plus the 700 randomly selected non-care home genomes (as described elsewhere in the paper). This new figure demonstrates that care home infections are intermixed across the phylogenetic tree with non-care home infections. This is consistent with the virus passing between care home and non-care home settings, rather than all the care home sequences being more closely related to each other than to other sequences. We have added this point to the figure legend and to the results:

“Consistent with this, care home and non-care home samples were intermixed across the phylogenetic tree (Figure 6A), suggesting viral transmission could pass between care homes and non-care home settings.”

The above sentence also means Figure 6 panel A is now discussed further in the manuscript than it was previously.

We have added a new table (Table 6) providing more detailed epidemiology on the COVID-19 outbreaks for the 10 care homes with the largest number of cases. These data indicate that, “while care homes frequently had more than one introduction of the virus among residents (i.e. >1 cluster), there was typically a single dominant cluster responsible for the majority of cases within each care home,” now in the Results section. The same trend is seen across all care homes with >3 samples. This is an interesting observation, and we thank the reviewers for prompting us.

2) Did the sequencing include the use of negative controls to detect possible contamination? Contamination is a common issue when using the ARTIC primers due to the many cycles of PCR amplification, and I have a hard time accepting sequencing data produced using this method without explicit description of the contamination controls or clean lab practices used. This information would be very helpful to see in the Materials and methods and would help promote careful validation of sequencing data.

The reviewer’s comment that, “contamination is a common issue when using the ARTIC primers due to the many cycles of PCR amplification…” is not entirely correct – any process that requires PCR is potentially sensitive to contamination, this is not a unique problem with “ARTIC primers”.

The workflow for nanopore sequencing performed in the Goodfellow lab at the University of Cambridge has been developed following years of field testing and validation. It relies on a directional sample flow as used in a diagnostic laboratory which includes separated pre- and post-PCR areas, with dedicated equipment for each stage of the process. All steps are performed in PCR cabinets which are cleaned using DNA removal solutions and a UV decontamination cycle run after each batch. All sequencing batches included at least one water negative control carried over from the reverse-transcription step. Mapped reads were assessed in real-time during sequencing with RAMPART (https://github.com/artic-network/rampart) and all data from batches containing a contaminated negative control were discarded before sequence assembly. This information is now included in Materials and methods.

The WSI sequencing workflow also uses negative controls and the pass rate to date related to negative controls is 90%. Sequencing read counts are considered after a clipping and minimum alignment length filtering step (corresponding to data which is used to create consensus sequence or variant calls). Such read counts for the samples analysed in this study were typically in the millions (median: 4,497,543). If such read counts for the corresponding negative controls are >100 then the samples are currently failed. This QC procedure was introduced for samples analysed on or after the 18th of April. Of the 1,007 samples analysed in this study sequenced at WSI (503 care home residents and 504 non-care home residents), 749 were sequenced once this workflow was established, 242 were sequenced before this but had a negative control and 16 did not have a negative control. If we apply the current criteria then 38 of these earlier samples would have failed (38/1400 = 2.7% of the analysed samples). 26 of these 38 samples are non-care home samples and 12 are from care homes. Of the 12 care home samples (12/700 = 1.7% total care home genomes analysed), 1 belongs to one of the "top 10" care homes with the largest number of genomes, care home CARE0063, which comprises a single cluster of 12 genomes using the *transcluster* algorithm, described in main text. Thus, the main result of our genomic cluster analysis (that multiple introductions are often observed in care homes, but typically a single dominant cluster causes most of the cases) would not be altered by the small number of early genomes included that would now be excluded by current criteria. This information has been added to the Materials and methods.

3) Similarly, did the authors investigate the ONT/Illumina sequence pair that had two SNPs between the sequences? There is evidence that ONT sequencing in particular has biases at particular areas within the SARS-CoV-2 genome (e.g., https://virological.org/t/issues-with-sars-cov-2-sequencing-data/473, https://www.medrxiv.org/content/10.1101/2020.08.13.20174136v2), and it would be interesting to know if this discrepancy occurs at one of these regions, which would allow the authors to explain and possibly even correct the error. If the error cannot be easily explained, the authors should comment on why they don't think their analysis, which is based heavily on single-SNP differences between sequences, could be affected by whatever might be causing this discrepancy.

We have looked into this pair of samples further. The two SNPs are identified as C1884T and C16351T. Neither of these are included in the list of problematic sites included in the virological article mentioned by the reviewer (sites that are described as highly homoplasic and have no phylogenetic signal and/or low prevalence), and we state this in the manuscript with this reference. In our previous study (Meredith et al., 2020), out of 14 sample pairs sequenced both by Illumina at WSI and nanopore in the University of Cambridge, there were zero SNP differences at positions where both sequences made a call.

We have a new paragraph in the Materials and methods outlining reasons why pairs of sequences from the same patient may have SNP differences, and link to a formal comparison of Oxford Nanopore Technology (ONT) vs Illumina sequencing which demonstrated highly accurate SARS-CoV-2 consensus sequences (https://www.biorxiv.org/content/10.1101/2020.08.04.236893v1.full). We conclude, “Given this degree of consensus sequence accuracy, and because transcluster uses a transmission probability cut-off based on integrating pairwise SNP and temporal differences (rather than relying solely on a strict SNP cut-off), limited sequencing noise is unlikely to have a substantial impact on the clusters identified.”

4) Overall the manuscript is fairly dense with a substantial amount of results and information provided in the text which would be better summarized as tables. For example, the authors state that 7,406 samples from 6,600 individuals were identified, but only 18% are from care homes? If so, is the 18% that was primarily analyzed in the manuscript? Or is this 18% only residents (and if so, are the care givers omitted?). Further, there are many values without percentages presented and it is difficult to follow which samples were considered in each section. For example, in the next section they report that 7% are from a hospital, but are these a subset of the first section? In general, it is difficult to follow how the samples/information in each section relate (or do not relate) to subsequent sections and exactly which samples (including the numbers and percentages) were analyzed.

We have added a more detailed legend to Table 1, which hopefully clarifies the samples analysed:

“The total sample set for this study comprised 6,600 individuals. Of these, care home residency status could be established for 6,413 (97.2%). 1,167/6,413 (18.2%) individuals were identified as being care home residents, of which 700/1,167 (60.0%) had genomic data available that passed quality control filtering and were used for identifying care home clusters using the transcluster algorithm (described in Materials and methods and main text). The subset of individuals (464/6,600, 7.03%) that were tested at Cambridge University Hospitals (CUH) had richer metadata available and were used for analysing intensive care unit (ICU) admissions and 30-day mortality after first positive test, shown here…”

When referring to “care home residents” from the study without qualification, we are referring to the 1,167 care home residents identified in the study. Of these, 700 individuals had genomic data available and these are generally referred to as, “care home residents with genomic data”, unless it is obvious from the context. We have ensured each paragraph of the Results section includes a description of what sample set is being used for that analysis, e.g. the subset of samples tested at Cambridge University Hospitals is clearly sign-posted:

“464 / 6,600 (7%) individuals with positive COVID-19 tests were patients tested at Cambridge University Hospitals. We had access to richer metadata for this subset of patients via the hospital electronic records system.”

The 464 individuals tested at Cambridge University Hospitals (CUH) are a subset of the total 6,600 individuals in the study.

Other examples:

“Genome sequence data were available for 700 / 1,167 (60.0%) care home residents from 292 care homes (Figure 2—figure supplement 2). […] Links between care homes and hospitals were investigated for the 700 care home residents with genomic data available.[…]Potential transmission networks involving care home residents and healthcare workers (HCW) were investigated for people tested at CUH (HCW data were not available outside of CUH). This analysis comprised 54 care home residents tested at CUH and 76 HCW with genomic data available”.

Regarding the density of numerical data presented in prose, we have cut out details that can be found in tables in several places. We have added a new table (Table 7) that summarises most of the numerical data presented in the “Links between care homes and hospitals” section, allowing us to cut back some of the counts listed in main text there.

5) The cluster results should provide more background and clarification. For example, it appears that only 60% of the data was analyzed in this section however it is not clear how this was decided. The authors should provide more information about the transcluster method since it is a newer method. This can help put the results in context, because currently is it not clear if the analysis considers all of the clusters together or only within cluster transmission. In addition, the authors state that 8 SNPs separate each care home, however they do not provide any indication if this suggest very close transmission amongst care homes. Additional lineage information (including statistics) would be helpful to put the data into context.

We have added substantially to the background of *transcluster* in Materials and methods, beginning with paragraph:

“Clusters were produced using an implementation of the transcluster algorithm (Stimson et al., 2019; Tonkin-Hill, 2020). Instead of targeting the number of SNPs separating two genomes, the transcluster algorithm proposes a probabilistic alternative which estimates the number of intermediate transmission events separating two sampled genomes. The method takes into account both genetic SNP distance as well as the time at which each sample was taken. The approach models both the SNP distance and the number of intermediate hosts as a Poisson process. Using a predefined evolutionary rate as well as an estimate of the generation time (the time between transmission events), the method infers the distribution of the number of intermediate hosts separating two samples.”

We go on to describe the mathematics of *transcluster* in further detail.

Re: the selection of 700 genomes; this was all care home residents with genomic data available for the analysis. We have clarified in the legend to Table 1 and at the start of each Results section exactly which samples were used for each analysis. In this case:

“Genome sequence data were available for 700 / 1,167 (60.0%) care home residents from 292 care homes (Figure 2—figure supplement 2).”

SARS-CoV-2 has low overall genetic diversity, so a median of 8 SNPs separating pairwise comparisons in a geographically contained area is quite typical and does not suggest care home genomes overall were especially low in diversity when compared against each-other. We put this into the context of the East of England region as a whole:

“There was a median of 8 single nucleotide polymorphisms (SNPs) separating care home genomes, compared to 9 for randomly selected non-care home samples (P=0.95, Wilcoxon rank sum test) (Figure 6—figure supplement 1), similar to the EoE region described previously (Meredith et al., 2020).”

We provide as much additional information on lineages as can be inferred from the available data:

“With ongoing viral evolution, descendent lineages of B.1 and B.1.1 also rose in frequency and were commonly found in England during the relevant time period. This suggests that the SARS-CoV-2 lineages circulating in care homes were similar to those found across the EoE outside of care homes… No new viral lineages from outside the UK were observed, which may reflect the success of travel restrictions in limiting introductions of new lineages into the general population.”

The main points we wish to make here are that the lineages inside and outside of care homes were similar, and similar to the East of England (and indeed Europe) as a whole. There were not specific “care home lineages” circulating separately from the non-care home wider community. We have added a new table (Table 5), which explicitly compared the frequency of lineage B.1.1 in early vs late time periods, for care home vs non-care home samples. We have not elaborated on other lineages (which would be complex to appreciate in a table and statistically challenging to compare) as we do not make strong claims on changes in particular lineage frequencies. We provide regional context in the references e.g. see Figure 2 of Elm et al. demonstrating the rise in frequency of lineage B.1.1 across Europe over the same time period (https://www.ncbi.nlm.nih.gov/pmc/articles/PMC7427299/).

Further, for outbreaks identified, information should be provided on the time these samples were collected.

We do not disclose the dates that care home outbreaks occurred/ samples were collected because of the risk of deductive disclosure: i.e. the risk that the combination of different anonymised data (e.g. number of cases for the care home, dates of sampling, age profile of residents etc) could be used to de-anonymise individuals. This concern is highlighted in the supplementary table of COG-UK IDs for the analysed genomes. If the reviewer is referring to the time of day the samples were collected, we do not have access to those data.

Further, the sentence "We investigated the role of genomics in defining care home clusters by repeating the transcluster algorithm using the same parameters as for the main analysis but assuming all genomes were identical " is not clear enough. What authors mean by "the role of genomics?".

We have explained this further in the same paragraph:

“The contribution made by genomic data in defining care home clusters was quantified. Without genomic data (or access to more detailed epidemiology such as accommodation sub-structuring within care homes), clustering can only be based on temporal differences between cases. For example, if two groups of COVID-19 cases occur several months apart within a care home they could be inferred to have resulted from (at least) two separate introductions. However, this method cannot account for multiple introductions occurring around the same time, as may happen when community transmission is high. To quantify the impact made by adding genomic data, which can distinguish between genetically dissimilar viruses introduced at similar times, the transcluster algorithm was repeated using the same parameters as for the main analysis but assuming all genomes were identical.”

Finally, a figure/table with SNPs, days separating and number of individuals per cluster would helpful since there is currently limited investigation provided. There is limited depth provided when exploring these clusters – for example the authors discuss a paramedic in cluster CARE0063 but they do not provide information of relatedness (SNP distance), how this sample related to the other samples in time, or any other information. In addition, if directionality cannot be inferred, this should be addressed within the limitations in the Discussion.

We have added a new table (Table 6) that shows a breakdown of epidemiological information for each of the 10 care homes with the largest number of genome samples. The pairwise SNP difference distributions for residents within these care homes is displayed in Figure 6B.

Our method of identifying clusters does not use a SNP difference cut-off, which is why we do not report SNP distance; instead, we use the *transcluster* algorithm, which integrates both SNP difference and date between sampling (used as a proxy for serial interval). We report that links for the HCW and care home residents shown in Figure 7B are in the top 1.1% of all pairwise transmission probabilities inferred using the *transcluster* algorithm, and the figure itself indicates the individual pairwise transmission probabilities based on their colour, as shown with the figure key.

The overall distributions of pairwise SNP and date differences within the clusters defined by *transcluster* are shown in supplements to Figure 7: Supplement 1 shows the distribution of pairwise transmission probabilities imputed by the *transcluster* algorithm (and the cut-off used to define clusters in our analysis); Supplement 2 shows how the number of clusters changes as the cut-off is changed (showing the cut-off used in our analysis); Supplement 3 shows a pairwise SNP difference histogram for samples within clusters; and Supplement 4 shows a pairwise date difference histogram for samples within clusters. We have added mentions for supplements 3 and 4 to the main text to highlight these.

We have provided more background on the *transcluster* algorithm in Materials and methods, which hopefully makes this methodology clearer.

Re: directionality of transmission – we have added this as a limitation in the Discussion, as suggested by the reviewer:

“Directionality of person-to-person transmission cannot be inferred from the transcluster algorithm. Inferring the likelihood of transmission direction between pairs of individuals requires integration with multiple forms of epidemiological data, yielding a probabilistic estimate (Illingworth et al., 2020).”

[Editors' note: further revisions were suggested prior to acceptance, as described below.]

Essential Revisions:1) Timing of samples: Information is needed to put the timing of samples into context – given that the method is highly reliant on this information. For example, were distinct samples taken a substantial period of time apart could possibly be part of the same transmission chain if not all cases were sampled? For example, dates by epi week or month/year are often allowed by IRBs. Or perhaps the authors could add some summary statistics to Table 6, e.g. span of time between samples in the care home and mean/spread of days between samples? This would be aggregate and could not be connected to a specific sample. Without additional information, the work is not transparent to the level that it can be properly evaluated. Additionally, we may have missed it but we don't think the authors specified the generation time they are using when running transcluster? Details on this and the uncertainty around this estimate would help, especially when compared to even relative/aggregate timing information (and a few more details on sampling).

We have added 2 columns detailing date information to Table 6. These columns show the date range in days (i.e. days from first sample to last sample date) for each care home and for each cluster within each care home. The date range for each care home is typically larger than the date range for clusters within care homes, except for single-cluster care homes like CARE0314 where the date range is already small for the care home as a whole. This is consistent with *transcluster* identifying groups of cases occurring closer together in time. It is also interesting to note cases like CARE0263, in which all 12 residents tested positive within 3 days of each-other, but these were in fact three separate clusters (one dominant cluster of 9 cases, one cluster of 2 cases and a single separate case), consistent with the three clusters that can be seen in the phylogeny shown in Figure 6A, i.e. *transcluster* is yielding both temporally linked cases and genetically linked cases, as expected. Without the genomic data, the three clusters in CARE0263 would have been impossible to distinguish. We have included this as an illustrative case in the Results.

The reviewer asks for aggregate data on time difference between samples within each cluster. A histogram of pairwise date differences between samples within each cluster is shown in Figure 7—figure supplement 5. We also describe these data in Results:

“Within each cluster, 673 / 775 (86.8%) of pairwise links had zero or 1 pairwise SNP differences (maximum 4), and 756 / 775 (97.5%) were sampled <14 days apart (maximum 22 days) (Figure 7—figure supplements 4-5).”

We have added Figure 7—figure supplement 6, which shows the median and interquartile range for pairwise date differences between all samples within each cluster, arranged from lowest to highest median date difference.

In addition, we have added an analysis of differences between sampling dates from first to last case for care homes versus clusters across the dataset:

“Clusters had a tighter distribution of sampling dates than for the total cases within each care home, as expected. For the 170 care homes with 2 or more cases with genomic data, there was a median of 9 (IQR: 4 – 15) days from the first case to the last case within each care home, compared with a median of zero (IQR 0-5) days from the first case to the last case of each cluster (P < 10^-5^, Wilcoxon rank sum test).”

Lastly, we have added sampling date to the list of COG-UK and GISAID IDs for samples analysed in the study to the supplementary materials.

Re: Generation time – we approximated the generation time by the serial interval, as is common in transmission studies (https://www.medrxiv.org/content/10.1101/2020.09.18.20197210v1), but recognise this was not clear in the Materials and methods. Alternative strategies for estimating the generation time have also led to similar estimates (https://www.eurosurveillance.org/content/10.2807/1560-7917.ES.2020.25.17.2000257). We have updated the corresponding sentences to read:

“The implementation of transcluster assumed a viral mutation rate of 1e-3 substitutions/site/year (Fauver et al., 2020) and generation time of five days, approximated by previous estimates of the serial interval of SARS-CoV-2 (He et al., 2020; Zhang et al., 2020). Days between first positive sampling date for pairs of individuals was used as a proxy for generation time.”

For example, we would be quite convinced that two samples taken 1-2 months apart could be ruled out to be separate introductions if there were no confirmed cases between them and the generation time was a few days with a small confidence interval.

We presume that the reviewer meant to say, “we would be quite convinced that two samples taken 1-2 months apart could be inferred to be separate introductions if there were no confirmed cases between them.”? If so, we agree that cases that occurred this far apart in time with no cases in between could be assumed to be separate introductions. The pairwise date difference between samples within each cluster is shown in Figure 7—figure supplement 5; most cases within each cluster are <14 days apart, with a maximum dispersion of up to 22 days. The distributions of differences in pairwise sampling dates between samples within each cluster are also shown in Figure 7—figure supplement 6. The *transcluster* clustering method is consistent with the reviewer’s intuitions.

2) Sampling bias: The revision does not adequate address the issues of sampling bias. For example, what was the depth of sampling in each care home? Were all positive cases sequenced? Only a proportion? What sampling bias (and subsequent impact on the results) may arise from the sampling strategies used? For example, the text states that: there is genomic information for "for 700 / 1,167 (60.0%) care home residents from 292 care homes" but only only 10 of these care homes were analyzed (102/700 total samples). It is not clear why all of the samples were not included and should be justified or these additional samples should be included in the analysis. This was included in the original reviews, however was not adequately addressed in the revision.

Every sample with an available genome at the time the analysis was run that passed sequencing quality control measures was included in the analysis. There were no exclusion criteria for genomic analysis other than the sequence quality control criteria described in Materials and methods. We attempted to sequence every sample tested in the Cambridge PHE diagnostic laboratory. The fact that genomic data were not available for 40% of care home residents is a limitation described in the Discussion, with reasons listed:

“Viral sequence data were not available for 40% of care home residents, as a result of missing samples, mismatches between sequences and metadata, genomes not passing quality control filtering using a stringent threshold (<10% missing calls), or sequences being unavailable at the time of data extraction.”

We have added to the Materials and methods stating the aim was to sequence all positive samples from the diagnostic laboratory, and list reasons why genomic sampling was incomplete:

“The study aimed to sequence all samples which tested SARS-CoV-2 PCR positive at the CMPHL during the study period. Sequencing of every positive diagnostic sample could not be performed, however, for the following reasons: (i) sample unavailability (e.g. diagnostic samples being lost or discarded before they could be collected by the sequencing team); (ii) labelling errors when assigning sequencing codes (which resulted in specimens being discarded); or (iii) metadata mismatches (if the sample did not match to a metadata record downloaded from the hospital electronic patient records system).”

There is no reason to believe that these factors were biased in any particular way. Moreover, 60% SARS-CoV-2 genomic coverage for all care home residents tested in our diagnostic laboratory is one of the highest rates of sequencing coverage for care home residents anywhere in the world.

We have re-written the caption for Figure 1 (the study flow diagram) to explicitly state how many samples were sequenced by nanopore and Illumina technologies:

“Out of 1,297 samples from 1,167 care home residents, 286 samples were assigned for nanopore sequencing on site and 833 samples for sequencing at the Wellcome Sanger Institute (WSI). Of these, 258 and 533 sequences were available and downloaded from the MRC-CLIMB server at the time of running the analysis, respectively. Of these available genomes, 224 and 522 passed sequencing quality control thresholds (described in Materials and methods), respectively. This yielded the final analysis set of 700 high-coverage genomes from care home residents (representing 292 care homes): 197 genomes sequenced on site by nanopore and 503 sequences at WSI by Illumina.”

All 700 genomes passing QC from care home residents were analysed in the *transcluster* analysis used to define care home clusters – none were excluded. We chose to visualise the “top 10” care homes with the largest number of genomes available in Figure 7 because we thought that displaying transmission networks for hundreds of care homes in a single, large figure would be difficult for a reader to appreciate. However, we have now added Figure 7—figure supplement 1, which shows the same network diagrams for every care home in the study with 2 or more genomes available. Care homes with only 1 genome available in the dataset are not displayed as the transmission network would consist of a single point.

One important finding of the cluster analysis is that care homes may have had multiple introductions among residents, but frequently a single cluster was responsible for the majority of cases (consistent with a substantial role of within-care home transmission driving the majority of care home cases). This is described for both the “top 10” care homes and aggregated for the entire dataset, quoting the relevant section of the previous manuscript version below:

“Of the 90 / 292 (30.8%) care homes with three or more residents with genomic data (comprising 418 / 700 (59.7%) care home residents with genomic data), 74 / 90 (82.2%) had a dominant cluster responsible for >50% of all cases in the care home.”

This analysis was limited to care homes with 3 or more cases with genomic data because 3 is the minimum number in which 2 clusters could be present and one be “dominant” (i.e. represent 2/3 cases). This still includes the majority (59.7%) of care home residents with genomic data. If the analysis is repeated for all care homes with 2 or more cases we still find the majority of care homes have one cluster comprising >50% of samples: Of the 170 / 292 (58.2%) care homes with 2 or more residents with genomic data available (comprising 578 / 700 (82.6%) care home residents with genomic data), 111 / 170 (65.3%) had a dominant cluster responsible for >50% of all cases in the care home. We have now added both numbers to the main text for transparency.

Regarding “sampling bias” from the point of sample collection (as opposed to which samples were sequenced/ analysed after testing positive), we agree that opportunistic sampling for genomic epidemiology carries a risk of sampling bias, and studies such as this must pay careful attention to this and consider how/whether it could affect their main conclusions. We describe this in the Discussion:

“…the nature of diagnostic testing sites changed during the study period as regional hospitals developed their own in-house testing capacity and community testing laboratories were set up. “Pillar 2” testing in the UK was outsourced to high-throughput laboratories during April 2020 and performed an increasing proportion of community testing. It is possible that some care home residents from the same care home could have been tested through different routes, with symptomatic cases more likely to be tested in “Pillar 1” via the CMPHL (and included in this dataset), and asymptomatic screening occurring more via the Pillar 2 laboratories.”

However, we feel this issue does not detract from our fundamental conclusions on clustering within care homes:

“…most care homes in EoE only began systematic screening after the end of our study following the introduction of the UK care home testing portal on 11th May 2020. Moreover, the transcluster algorithm allows for “missing links” within a cluster (the threshold used assumed a >15% probability of infections being linked within <2 intermediate hosts), reducing the impact of missing care home cases on defined clusters.”

And finally, we warn that:

“The changing profile of COVID-19 testing in the UK between March and May 2020 should therefore be factored into all interpretations of COVID-19 epidemiology from that period.”

3) Provide additional detail on the phylogeny: Additional information is needed in the text to ensure that the conclusions based on SNP differences are supported by the phylogeny. For example, seeing the tree Figure 6A in more detail (e.g. a zoom in panel), would allow the reader to see if non-care home sequences fall within the colored clusters or not. This would also help address the issues of sampling bias (care and non-care home individuals) and the subsequent implications. For example, the authors say "Samples from the ten care homes with the 1180 largest number of genomes are highlighted by coloured circles on branch tips." : if I look at the first colored circle in cyan corresponding to CARE0151, based on Table 6 I would expect to see 7 samples, but I only count 5. It is overall difficult to deciphering the tree. The addition of more information in the figure legend would also be beneficial and avoid the reader from searching the information in the text. Nonetheless, it is surprising that the transcluster method, that defines clusters based in genetic information but also date of collection, identifies clusters with sequences that are scattered all over the tree like in the case of clusters CARE0151, CARE0277, CARE0061 or CARE0032. The fact that the clusters are heterogeneous is reflected in the pairwise SNP difference plot, and very clearly for CARE0277 that seems to have two sub-populations, but does not appear to be the case for CARE0151. These might need more in depth explanation despite the observation reported in “By contrast, several care homes were “polyphyletic”, with cases distributed across the phylogenetic tree and higher pairwise SNP difference counts between samples, consistent with multiple independent introductions of the virus among residents.”, for example.

We have added a magnified subtree for one branch of the phylogeny in Figure 6A, focusing on the monophyletic branch for care home CARE0314. This demonstrates these genomes are identical or 1-SNP different (consistent with the box plot in panel 6B). Two non-care home genomes are also present in this clade. We have stated that care home and non-care home genomes are intermixed in the tree, consistent with viral transmission occurring between care home and non-care home settings, which is expected.

Re: The number of coloured tips on the tree, some samples close together on the tree have coloured tips “on top of each other” so the number of circles visualised may be fewer than the total plotted. We have checked and this applies to CARE0151.

Re: “heterogeneous clusters”. To be clear, Figure 6B shows pairwise SNP differences for the 10 care homes with the largest number of samples, not pairwise SNP differences within clusters defined by transcluster. Some care homes, such as CARE0314, are “monophyletic”, with low pairwise SNP difference count across all samples from that care home, and CARE0314 is identified as having a single cluster by *transcluster*. Other care homes, such as CARE0151, are “polyphyletic” and have higher pairwise SNP differences between all samples from that care home, and *transcluster* accordingly divides this care home into multiple clusters as shown in Figure 7A. ie. Figure 7 shows the clusters defined by *transcluster*, Figure 6 is showing a phylogenetic tree and pairwise SNP differences within care homes, not clusters. We have made this clearer in the legend to Figure 6:

“Clusters within each care home were defined using integrated genomic and temporal data using the transcluster algorithm and are shown in Figure 7.”

The pairwise SNP difference between samples within each cluster is shown in Figure 7—figure supplement 4, and is also summarised in main text:

“Within each cluster, 673 / 775 (86.8%) of pairwise links had zero or 1 pairwise SNP differences (maximum 4), and 756 / 775 (97.5%) were sampled <14 days apart (maximum 22 days) (Figure 7—figure supplements 4-5).”

The phylogenetic tree and pairwise SNP difference plot shown in Figure 6 illustrate the general principle of the distinction between polyphyletic and monophyletic care home cases and demonstrate that care home genomes are phylogenetically intermixed with non-care home genomes. The paper goes on to define care home clusters formally using *transcluster*, integrating the genomic data with temporal data. All conclusions drawn on clustering within care homes are based on the *transcluster* analysis, which is described at length in the two paragraphs detailed in Materials and methods and visualised in Figure 7.

[Editors' note: further revisions were suggested prior to acceptance, as described below.]

Unfortunately the Editors and Reviewers felt that the manuscript is not acceptable for eLife as it has not reached the level of clarity needed to allow for the key conclusions to be evaluated and for someone else to replicate these results.

We recognise the importance of reproducibility in the scientific method. However, this must be balanced against the ethical requirement for study participant confidentiality in line with the approved ethics for the study. COG-UK has ethical approval to publicly release limited sample metadata including the person’s age, sex and county of residence and the date of sample collection. Releasing additional metadata linked to COG-UK IDs, such as whether the person was a care home resident and their relationship to other individuals via anonymised care home codes, would risk deductive disclosure. For example, it may be possible to work out who individuals are based on knowing 3 people from the same care home in Cambridgeshire who tested positive on particular dates, and knowing their age and sex. This breach of confidentiality would be unethical and violate the ethical approvals in place to do the study.

We refer to the General Medical Council (GMC, the regulatory body for medical doctors practicing in the UK) guidance on confidentiality (https://www.gmc-uk.org/ethical-guidance/ethical-guidance-for-doctors/confidentiality), section “Using and disclosing patient information for secondary purposes”, pp 37-39. The section on anonymised information states (our emphasis):

“The Information Commissioner’s Office anonymisation code of practice (ICO code) considers data to be anonymised if it does not itself identify any individual, and if it is unlikely to allow any individual to be identified through its combination with other data. Simply removing the patient’s name, age, address or other personal identifiers is unlikely to be enough to anonymise information to this standard.”

This is based on the UK Information Commissioner's Office (ICO) guidance, available here: https://ico.org.uk/media/1061/anonymisation-code.pdf

We have discussed with senior members of the COG-UK consortium (several of whom are co-authors on this manuscript) who have confirmed that care home residency status including anonymised care home codes are on the COG-UK restricted data list and cannot be released publicly.

To address the issue of reproducibility we have generated a version of the dataset with anonymised sample names – i.e. the genetic distance matrix and linked anonymised metadata required to run the transcluster analysis but without samples being linked to their COG-UK sequence codes. This anonymised dataset includes the same anonymised care home codes as used in the paper and all code so the results are fully reproducible. The data and code are publicly available via GitHub at https://github.com/gtonkinhill/SC2-care-homes-anonymised.

In particular, the key piece of information that is missing is the amount of time between samples and clusters within the same care home that are needed to understand how the genetic data is being used to determine the clusters. While Table 6 is helpful, it still does not provide enough detail about the timing of infections – in particular the time between clusters within a care home and the mean time/distribution of time between samples within and between the care homes. These data are needed to interpret the results. Given limitations to protect the privacy of participants, information by care home – including clusters identified, dates associated with these clusters, and sampling timeframes – is acceptable as an alternative to including the care home information per sample.

We have added two large supplementary tables that provide all of the requested data, expanded the description of date differences in the Results, and added another supplementary figure comparing the date range distribution for care homes and clusters.

The two new tables in Supplementary Materials are, “Sampling date ranges for care home residents with genomic data: by care home”, and, “Sampling date ranges for care home residents with genomic data: by cluster defined by the *transcluster* algorithm”. These provide the sampling dates for the first and last samples for every care home and cluster in the full dataset, respectively. This should provide all of the information the reviewers ask for above, including number of clusters from each care home, number of samples within each care home and cluster, and the sample date range for every care home and cluster in the study.

Summary data on within-cluster sampling date distributions is shown in Figure 7—figure supplement 7, and the time ranges from first to last samples within clusters and within care homes is described in the Results. We have added Figure 7—figure supplement 6, showing boxplot distributions of date ranges (from first to last sample dates) for care homes vs clusters.

Further, since the authors make a point to distinguish between monophyletic and polyphyletic cluster, a full tree should be included in the supplementary information (while the added zoom is a nice addition, it is lacking since it is just about one cluster).

We have added Figure 6—figure supplement 1, which is a phylogenetic tree for all samples in the study with available genomic data that passes sequencing quality control (total N = 4,445 samples). The figure is produced in high resolution, so that the interested reader can zoom in and view every branch of the tree.

Finally, the data sharing agreement does not meet the standards required for eLife. That is, simply stating that others may "discuss the process of signing a data sharing agreement" appears particularly subjective. The authors need to have a full data package available to anyone who requests it – or any limitations on providing the full data need to be specified. Eg: if it is limited to academic (non-commercial) study / people in a particular geographical jurisdiction / a confidentiality agreement is required – a draft of this agreement should be provided and a clear statement on what reasons for declining would be. These raw data need to be accessible to allow attempts at replication as an important part of the scientific process.

As above, the anonymised dataset and code now available on GitHub allows full reproducibility of the clustering results generated using the *transcluster* algorithm, without requiring any data sharing agreement.

If a researcher requires access to restricted metadata (including care home residency status) linked to the COG-UK sequence codes, then this will require a formal data sharing agreement with the COG-UK Consortium. Access to patient outcome information for patients treated at Cambridge University Hospitals NHS Foundation Trust (CUH) requires a data sharing agreement with CUH. Data will only be shared for public health and research purposes, not for commercial enterprise, and only to individuals working at reputable research and public health institutions for which data security can be assured. Should this be required researchers should contact the study corresponding authors in the first instance.

[Editors' note: we include below the reviews that the authors received from another journal, along with the authors’ responses.]

Reviewer 1 (stats)I will focus on methods and reporting. This is an interesting and valuable piece of reseach but I have some reservations.The process to identify care home residents is not perfect but it is comprehensive. Some cases may be missed but I would not expect them to be many.Major1) It wasn't clear to me if testing occuring for each care home ending up int he authors' database, for the time period of interest, would be available. In other words, it is possible for some of the testing taking place being in the database (for a particular care home – except those not appearing in the data at all), but not all? If not all tests are available, this would be a severe limitation. The first implication that this may be the case appears in the Results section, as percentage of all cases compared to those in the whole of the East. But that doesn't discuss care homes specifically (which is of course hard to do without doing the research for those other cases, but in the context of consistent submissions by care homes, it is an important point).

We capture around half of all COVID-19 cases that occurred in the East of England over the study period, the remaining positive cases being sampled at other sites. We have expanded our description of where the samples included in the study derive from in the Results:

“The samples were tested at the Public Health England (PHE) Clinical Microbiology and Public Health Laboratory (CMPHL) in Cambridge, which receives samples from across the East of England. Positive cases came from 37 submitting organisations including regional hospital laboratories and community-based testing services (Supplementary file 1). The proportion of samples coming from different sources changed over the study period (Figure 1—figure supplement 2). This likely reflects a combination of regional hospitals establishing their own testing facilities, increasing availability of community testing in the UK, and the implementation of national policies that increased the scope of care home testing (Figure 1—figure supplement 3). Overall, the study population included almost half of the COVID-19 cases diagnosed in the EoE at this time (Public Health England, 2020a), with the remainder being tested at other laboratory sites.”

Figure 1—figure supplement 2 and Supplementary file 1 provide breakdowns of which sites submitted samples and were included in the study.

Specifically addressing the reviewer’s question of whether one care home could have submitted some samples to the Cambridge CMPHL (getting included in this dataset) and other samples to different testing sites (not included in this dataset), the answer is yes this is possible. As community testing expanded, more “screening” samples were sent to “Lighthouse Laboratories” (in the UK “Pillar 2”), while symptomatic cases were more likely to have been tested via Pillar 1 (the CMPHL). However, we do not believe this limitation is as “severe” as the reviewer suggests, as the *transcluster* algorithm we have used for defining clusters does allow for “missing links” connecting a transmission chain within a threshold number of intermediate hosts, based on the expected serial interval and mutation rate of the virus. We now address this issue explicitly as a limitation in our Discussion:

“We acknowledge several limitations to this study. First, we have not captured all of the COVID-19 cases from the East of England. Serology data indicate that 10.5% of all residents in care homes for people aged 65 and older in England had been infected with SARS-CoV-2 by early June, the majority of whom were asymptomatic (UK government, 2020c). […] The changing profile of COVID-19 testing in the UK between March and May 2020 should therefore be factored into all interpretations of COVID-19 epidemiology from that period.”

2) There is no clear justification for the selected time period. It is because of convenience becaue the data are available? Why not use the national portal data as well? Was it because it was the first outbreak within that time period? Anyway, some justification is needed.

The time period used spanned from the first positive case received in our laboratory (26^th^ February), to the 10^th^ May. This end-date was chosen because: (1) it captured the majority of the “first wave” of the epidemic in the East of England (as shown in the epidemic curves included); (2) Due to some delays in sequencing and genomic data becoming accessible after sample collection, the availability of genomes at the time data was pulled for analysis declined after that date; and (3) the national care home testing portal opened on 11^th^ May 2020, and this potentially could have introduced bias in population analyses as care homes may have undergone systematic screening. As described above, the distribution of testing was complexifying over April and May such that once systematic care home screening became more common, and the Lighthouse Laboratories were running, it becomes more likely that different samples from the same care home could be tested at different sites, making the picture from the Cambridge CMPHL less complete. We have made this more explicit in the Materials and methods:

“The 10^th^ May was selected as a study end-date because it encompassed the bulk of the “first wave” of the epidemic in the East of England. Furthermore, prior to the 11^th^ May 2020, systematic screening of all residents within care homes was much less common and testing primarily occurred where there was a suspicion of an outbreak; our strategy reduced risk of bias which would have been introduced had we included systematic screening.”

3) the Mann-Whitney U test (more commonly used name than Wilcocox rank-sum test) is appropriately used to compare the age of care home residents vs other cases. However, I struggle to see what the test tells us when comparing cases. How can that comparison be informative without the populations in each nursing home and each residential home? A more appropriate approch would be to model cases as count data, using a Poisson or negative binomial regression model with the care home as the unit of analysis.

In a Poisson regression model, increasing case numbers per care home was weakly associated with nursing homes relative to residential homes (odds ratio (OR) 1.21 (95% confidence interval (CI) 1.08 – 1.35), *P*=0.00132). However, we agree with the reviewer’s point that this needs to be interpreted in the context of the populations in each home. The CQC dataset includes number of beds registered with each care home. This could be a rough proxy for number of residents in each care home (though bed occupancy may not be 100%, and there may be some turnover of new patients into the beds over time). When using positive cases per CQC registered beds for each care home, there was a slightly *higher* positivity rate in residential homes than in nursing homes: median 0.063 (IQR 0.033 – 0.11) cases per bed vs median 0.048 (IQR 0.026 – 0.066), respectively (*P*=0.008322, Wilcoxon rank sum test). We are not sure of the significance or applicability of this observation, and it is tangential to the main narrative of the paper. We have therefore removed the comment on different case numbers between residential and nursing homes from the results.

4) Another issue is what is the genome sequence bringing into the epidemiology of COVID-19. The only thing I can see is whether the outbreak is monophyletic or polyphyletic. I am not sure a clear argument about the importance of this, from the public health point of view has been made – perhaps the authors need to make this clearer.

This is an important point; we have expanded on the role of genomics in the analysis and in addressing public health questions in several places.

We repeated the viral clustering algorithm assuming all genomes were identical; this effectively eliminates the contribution of genomics to the clustering so only temporal differences between samples are used to define clusters. This yielded 316 clusters, significantly fewer than the 409 yielded when genomics was included. This shows that without genomics, distinct transmission events occurring around the same time cannot be distinguished, so viruses are grouped incorrectly into the same clusters when they are actually more likely to be distinct introductions. This analysis has been added to the Results section describing the cluster work:

“We investigated the role of genomics in defining care home clusters by repeating the transcluster algorithm using the same parameters as for the main analysis but assuming all genomes were identical to each other. This yielded 316 clusters from the 700 residents across 292 care homes – 23% fewer than the 409 clusters yielded when incorporating genomics. This suggests that genomics makes a significant contribution to defining viral clusters; without genomic data, separate transmission events occurring around the same time in a care home cannot be readily distinguished (so cluster sizes are over-estimated and the number of separate viral introductions is under-estimated).”

We emphasise this point in the Discussion, e.g.:

“We defined viral clusters within each care home by integrating temporal and genetic differences between cases. This provides a “high resolution” picture of viral transmission; without genomic data, separate introductions of the virus occurring around the same time would be impossible to distinguish.”

And,

“incorporating genomic data is more accurate for excluding linked transmission than if only temporal data are available. Genomics can thus be used to “rule out” cases as being part of a linked cluster if the genetic difference is greater than would be expected given the viral mutation rate. This could be practically informative for care homes (along with other organisations at risk of COVID-19 outbreaks like factories (Middleton et al., 2020)) with implications for infection control procedures.”

An example application of integrating genomics into cluster definitions is to assess whether a care home has a single “outbreak” or unconnected transmission events. PHE currently defines an outbreak as 2 or more cases within the same care home within 14 days. Once an outbreak is declared there are significant infection control implications, such as closing to all non-essential visitors (currently for 28 days). This period resets each time a subsequent case occurs within that time period – so care homes can remain closed to visitors for extended periods, which is obviously difficult for the residents and their families. Genomics could be used to rule out certain cases as being part of a linked transmission cluster occurring within that time window, with implications for whether the care home continues to operate under its “outbreak” protocol for visitor restrictions.

We have also emphasised the key public health messages of the study more strongly in the Discussion – the need for strong infection prevention and control within care homes to limit transmission:

“These findings emphasise the importance of limiting viral transmission within care homes in order to prevent outbreaks. Given that SARS-CoV-2 is thought to be infectious before the onset of symptoms (He et al., 2020), isolating residents or staff when they develop symptoms is not sufficient to prevent within-care home spread once the virus has entered the care home. Certain measures may be required on an ongoing basis within care homes when there is sustained community transmission, even when no outbreak is suspected (at least until the morbidity and mortality of the virus in older people has been reduced substantially through vaccination or treatments). These may include use of appropriate Personal Protective Equipment (PPE) for staff and visitors (including visiting healthcare professionals and friends and family), rigorous hand hygiene, social distancing, and making use of larger, well-ventilated rooms for social interactions or socialising outdoors, providing that this is practical and safe (Jones et al., 2020). This is consistent with current national guidance for care homes in England (Public Health England, 2020b; UK government, 2020b). Face coverings for residents themselves when interacting socially in communal indoor areas could be considered, if acceptable to residents.”

Minor1) Quality of the Abstract is substandard. Even after going over it a few times, I struggle to understand the design, the samples and some of the reported findings. How were samples colelcted. How may samples per patient. Wouldn't it make sense to report positive patients rather than positive samples? why is the CHU subsample of 71 patients highlighted?

We have re-written the Abstract to emphasise the key points from the study, particularly the central public health point on the importance of limiting within-care home viral transmission

Reviewer 2This is a well written and informative manuscript which addresses transmission of SARS-CoV2 among a particularly vulnerable population. Overall, the authors are to be commended for their diligence, not just in their sequencing efforts, but in tracing and checking the , no doubt, highly complex world of CQC records. The methodology, both for sequencing and data analysis, is comprehensively described and the conclusions reached are well supported by the data. It highlights the potential for within nursing home and between nursing home transmission, and shows clearly the potential involvement of healthcare workers. The findings should have implications for infection control, and surveillance in nursing homes.

We thank the reviewer for their comments.

A couple of areas would welcome clarification:1) The genomes were sequenced using the nanopore and illumina platforms. Were a subset of samples cross checked on the other platform to ensure the same results? I might have missed this in the appendix, but would help ensure no disparity between different sequencing platforms.

Thank you for this comment, this is an important point. In our previous study (Meredith and Hamilton, *Lancet Infectious Diseases*, DOI: https://doi.org/10.1016/S1473-3099(20)30562-4), 14 genomes were sequenced on both platforms, and we found zero instances in which a different nucleotide was called between a pair of sequences. We have now checked our dataset from this study and identified 8 instances in which the same care home resident was sequenced on both Illumina and Nanopore technology. In 7 cases the pairs of sequences have zero SNP differences; in one case there are two SNP differences, both C vs T calls in different parts of the genome. We have added this analysis to the Materials and methods section, including the COG-UK IDs of the 8 pairs of genomes.

2) The study, due to testing, bases transmission modelling on symptomatic testing- perhaps a limitation of the study is that asymptomatic transmission couldnt be determined. If this was included, there may have been more within and between nursing home transmission accounting for some of the cases observed.

We agree with the reviewer as prior to 11^th^ May (when the UK national care home testing portal was introduced) care homes were unlikely to have been screening residents for COVID-19. We highlight in the Discussion the broader limitation that we have an incomplete picture of COVID-19 in the region, due to limitations on testing, asymptomatic/ pauci-symptomatic cases, some individuals getting tested at other sites not included in our study, and not all tests producing a genome that passed quality control filtering. We have also added a reference to the UK Vivaldi study, which estimated care home SARS-CoV-2 prevalence and found a high proportion of asymptomatic cases:

“We acknowledge several limitations to this study. First, we have not captured all of the COVID-19 cases from the East of England. Serology data indicate that 10.5% of all residents in care homes for people aged 65 and older in England had been infected with SARS-CoV-2 by early June, the majority of whom were asymptomatic (UK government, 2020c). The Cambridge CMPHL did not receive all of the samples from the region, though based on national data we estimate we have captured around half of the COVID-19 cases reported from EoE during the study period. We did not have viral sequence data available for 40% of care home residents, as a result of missing samples, mismatches between sequences and metadata, genomes not passing quality control filtering using a stringent threshold (<10% missing calls), or sequences being unavailable at the time of data extraction. We may therefore have underestimated viral cluster sizes.”

However, one of the strengths of modelling clusters using *transcluster* is that we can, to some extent, account for “missing links” in a connected transmission cluster (whether because those links were asymptomatic and not tested, or the samples did not produce usable genomes, *etc*). *transcluster* allows for a probability of transmission within a set number of intermediate hosts – in our case we chose a relatively relaxed probability (>15%) of transmission occurring within 2 or fewer intermediate hosts. We describe this in the next paragraph after the above quotation in the Discussion:

“the transcluster algorithm allows for “missing links” within a cluster (our thresholds assumed a >15% probability of infections being linked within <2 intermediate hosts), reducing the impact of missing care home cases on defined clusters.”

Reviewer 3In this study, genomic epidemiology was used to investigate viral transmission dynamics among care homes residents and health care workers. It aimed at answering questions about the burden of care home associated COVID-19; patterns of SARS-CoV-2 spread between care home residents (single and multiple independent transmission networks); and the role of health care workers in the viral spread.Major:Figure 3A. In many points along the manuscript, some genomes are said to "cluster together on the phylogenetic tree", but the current plot of the phylogeny (Figure 3A) does not allow readers to inspect such clustering pattern. Please provide inset panels highlighting the referred clusters, including the statistical support of each clade, especially those suggested to represent single introductions. Please also consider adding a scale bar (in subs/site, or mutational units). It would be helpful if a full phylogeny could be provided as supplement, with labels and all the annotations suggested in this review.

We have re-produced this figure and now highlight the ten care homes with the largest sample numbers as coloured tips to the branch ends, rather than as the adjacent colour bar. Hopefully this makes the clustering easier to appreciate.

Appendix. As shown in the table, among the 6600 cases reported in the period of study, around 19% were linked to home care residents, while most of the remaining cases were associated with community acquired infections. If the proportion between “care home” X “non-care home” cases were nearly 20:80, why the proportion of genomes sequenced from each category was 50:50 (700 genomes each)? How the over-representation of ”care home resident” genomes (and consequent sub-representation of non-care home ones) could have impacted the phylogenetic and transmission network analysis?

We only sampled a matching set of non-care home genomes to get a sense of whether the predominant viral lineages seen in the care home residents was similar to non-care home residents (and to the UK/ Europe as a whole), and this is only for a single figure supplementary to the main narrative. We do not use this sample of non-care home genomes anywhere else in the analysis, including the cluster analysis (*transcluster*) or the phylogenetic tree, so it does not impact on these analyses. We now make that point explicitly clear in the Materials and methods. To formally compare care home and non-care home genomes, we agree multiple factors would need to be controlled for between the populations including the proportion of cases from different time periods, age, county locations, etc. However, as we do not rely on the non-care home samples for any heavy conclusions in this study, we do not feel this is necessary here.

Appendix. Please supplement the last table in the appendix with information about: GISAID or Genbank virus ID (“strain name”) and accession numbers (if submitted to other databases), date of collection, geographic location, and other relevant metadata associated with the genomes used in this study. This information is essential for the reproducibility of the results.

The consensus fasta sequences used in our analyses are publicly available and can be downloaded via COG-UK or GISAID websites. We have added virus name and GISAID accession numbers along with the COG-UK IDs for all analysed genomes to the Supplementary Materials. When linking GISAID IDs to their corresponding COG-UK IDs for this publication we identified two samples where the sequence we analysed was different to the publicly available sequence on COG-UK: CAMB-1ADAE8 and CAMB-1AEB6C, and there were no GISAID IDs for them. After some investigation we found this was due to low coverage in an initial run, which was uploaded to COG-UK and did not pass GISAID QC filtering, but the samples were subsequently re-sequenced with higher coverage, and this was used in our analysis. We are in the process of getting the re-sequenced (high coverage) versions uploaded to COG-UK, and then they will be assigned GISAID accession numbers. This should happen in time for publication.

The public databases like GISAID include metadata such as patient age, sex, country of sampling etc. However, for reasons of patient confidentiality and information governance, not all of the metadata used in our analysis can be released publicly. This is because of the risk that, when used in combination, certain metadata could de-anonymise individuals. We have added the following explanation to the supplementary table of analysed genomes:

“Sequences have associated public metadata (also available via the COG-UK website or GISAID), including patient age, sex, collection date (if available), and location to the level of UK county. However, not all of the metadata used in this study can be released publicly. COG-UK samples are sequenced under statutory powers granted to the UK Public Health Agencies. Matched patient data is securely released to the COG-UK consortium under a data sharing framework which strictly controls the handling of patient data. The status of individuals living in a care home and groups of such care home patients are both on the consortium restricted data list. This means that this data cannot be publicly released linked to sequencing identifiers, sampling date and UK counties. This is because of the risk of deductive disclosure. If a research scientist would like to repeat our analysis using these data fields, they should write to the corresponding authors to discuss the process of signing a data sharing agreement that will allow them to access the data securely.”

Minor:Figure 3. Please provide the number of genomes found in each bin on Panel B. To ease interpretation, consider colouring the branches/tips of the phylogeny according to the legend on panel A. If the same is done on panel 3B, that would be helpful.

We have followed the reviewer’s advice and coloured the tips of the tree by the care home, rather than using a colour bar. We have also added the number of genomes for each care home to the box plot (panel B), as suggested.

Could you please provide additional information as to why less then 7% of home care patients were admitted to ICUs, while 42.3% of then died? Did these deaths occur in the care homes, prior to hospitalization? This figure is especially striking when they are compared to those from non-care homes (21.4% and 17.3%, respectively). Were all deaths of non-care home residents associated with patients admitted to ICUs?

We have added the following addressing this point, with references, to the Discussion:

“A smaller proportion of care home residents were admitted to ICU compared with people who were not from care homes. What treatments a patient receives, including the invasive treatments provided in intensive care, are complex and individualised decisions based on risk-benefit assessments involving patients, their families and carers, and healthcare professionals (ICS, 2020; NICE, 2020). Of note, non-invasive respiratory support (such as continuous positive airway pressure, high-flow nasal oxygen therapy and non-invasive ventilation) are routinely provided outside ICU in many UK centres.”

Please also note that the mortality figures have slightly changed from original submission, (1) because we include a care home resident tested at CUH who was not included in the Cambridge COG-UK dataset (detailed in Materials and methods), bringing the total CUH care home residents from 71 to 72, and (2) because we now use mortality at 30 days post positive test, rather than mortality within the initial admission (thus including people who may have been discharged from hospital in a palliative setting who died at home or elsewhere such as in a nursing home). This is more consistent with the way mortality data is reported nationally.

It is not clear if all non-care home samples came from residents of EoE. Could you please clarify?

All samples and genomes in the study, including non-care home residents, came from the catchment area of tests performed at the Cambridge PHE Clinical Microbiology Laboratory, all of which are from the East of England. We have made this explicitly clear in the Materials and methods.

The manuscript made use of two distinct analyses to reveal potential clusters of cases: one used phylogenetics, and another used transmission networks. In both analyses, distinct elements are included: HCW, care house residents, and non-care house residents. It would be beneficial if nodes representing those elements could be highlighted in both graphs, so that they can be easily identified.

Both the phylogenetic and *transcluster* analyses identified clusters of residents within each care home. Neither analysis involved healthcare workers or non-care home residents, so these cannot be annotated in the phylogenetic tree of cluster analysis (every individual is a care home resident). However, we did a separate analysis focusing specifically on potential transmission links between healthcare workers and care home residents. The example discussed in main text, involving care homes CARE0063 and CARE0273, is shown in Figure 7B and the healthcare workers are marked with a different colour to the care home residents.

"1 pairwise SNP difference"

That sentence is “zero or 1”, so either “differences” (if referring to “zero SNP differences”) or “difference” (if referring to “1 SNP difference”) would be correct. We think “zero or 1 SNP differences” reads more fluently but this may be subjective.

Appendix, Page 35: The figure in this page shows CARE0273, which is not mentioned in the main manuscript. Would that be CARE173?

The figure is correct in labelling this CARE0273. That care home is mentioned in the section of the Results on the HCW-care home link analysis:

We also observed cases from a third care home, CARE0273, with possible transmission links to the same paramedics and carers involved in the CARE0063 cluster. These two care homes are within 1 kilometre of each-other and the cases cluster together on the phylogenetic tree, raising the possibility of shared transmission between them.

CARE0273 is not among the “top 10” care homes with the largest number of genomes and so not included in the figures showing transmission links between residents within the ten largest care homes.